# A model of tension-induced fiber growth predicts white matter organization during brain folding

Kara E. Garcia [1,2✉], Xiaojie Wang[3,4] & Christopher D. Kroenke [3,4]

The past decade has experienced renewed interest in the physical processes that fold the developing cerebral cortex. Biomechanical models and experiments suggest that growth of the cortex, outpacing growth of underlying subcortical tissue (prospective white matter), is sufficient to induce folding. However, current models do not explain the well-established links between white matter organization and fold morphology, nor do they consider subcortical remodeling that occurs during the period of folding. Here we propose a framework by which cortical folding may induce subcortical fiber growth and organization. Simulations incorporating stress-induced fiber elongation indicate that subcortical stresses resulting from folding are sufficient to induce stereotyped fiber organization beneath gyri and sulci. Model predictions are supported by high-resolution ex vivo diffusion tensor imaging of the developing rhesus macaque brain. Together, results provide support for the theory of cortical growth-induced folding and indicate that mechanical feedback plays a significant role in brain connectivity.

[1] Indiana University School of Medicine, Department of Radiology and Imaging Sciences, Evansville, IN 47715, USA. [2] Washington University in St. Louis, Department of Mechanical Engineering and Materials Science, St. Louis, MO 63130, USA. [3] Oregon Health and Science University, Division of Neuroscience, Oregon National Primate Research Center, Beaverton, OR 97006, USA. [4] Advanced Imaging Research Center, Oregon Health and Science University, Portland, OR 97239, USA. ✉email: karagarc@iu.edu

In gyrencephalic species such as the human, several lines of evidence support a link between the structure of brain folds and the connectivity of underlying white matter. In clinical disorders such as epilepsy, autism, bipolar disorder, and schizophrenia, differences in both the pattern of cortical folds and white matter organization have been reported[1–4]. In animal models, lesion experiments directed at specific axon fiber tracts have been shown to induce abnormal cortical morphology[5]. Furthermore, gyral and sulcal structure inherently influence the lengths of axons necessary to connect different regions of the cortex[6]. However, the biological and physical mechanisms linking white matter organization and cortical folding remain ill-defined (Fig. 1).

During the developmental period corresponding to folding, prospective white matter tissue—which includes the subplate and deeper fibrous layers[7]—undergoes notable changes. Directly beneath the cortex, the subplate transforms from a loose arrangement of cell bodies, radial glial scaffolds, and neuronal processes, into a tightly organized, axon-rich tissue[8]. Thickness of the subplate has been correlated to increasing gyrification as well as increasing complexity of cortico-cortical (association) fiber systems, with short association fibers emerging around the time of gyrification[8–10]. In the adult human brain, short association fibers that connect adjacent gyri (U-fibers) exhibit highly stereotyped organization with respect to fold morphology (Fig. 1d), and long association fibers (such as longitudinal fasciculi) are highly consistent across individuals[11,12]. Based on these observations, subcortical organization has been proposed to actively direct folding, either by tethering specific, highly connected regions[6,13] or by pushing the cortex outward to form gyri[8–10].

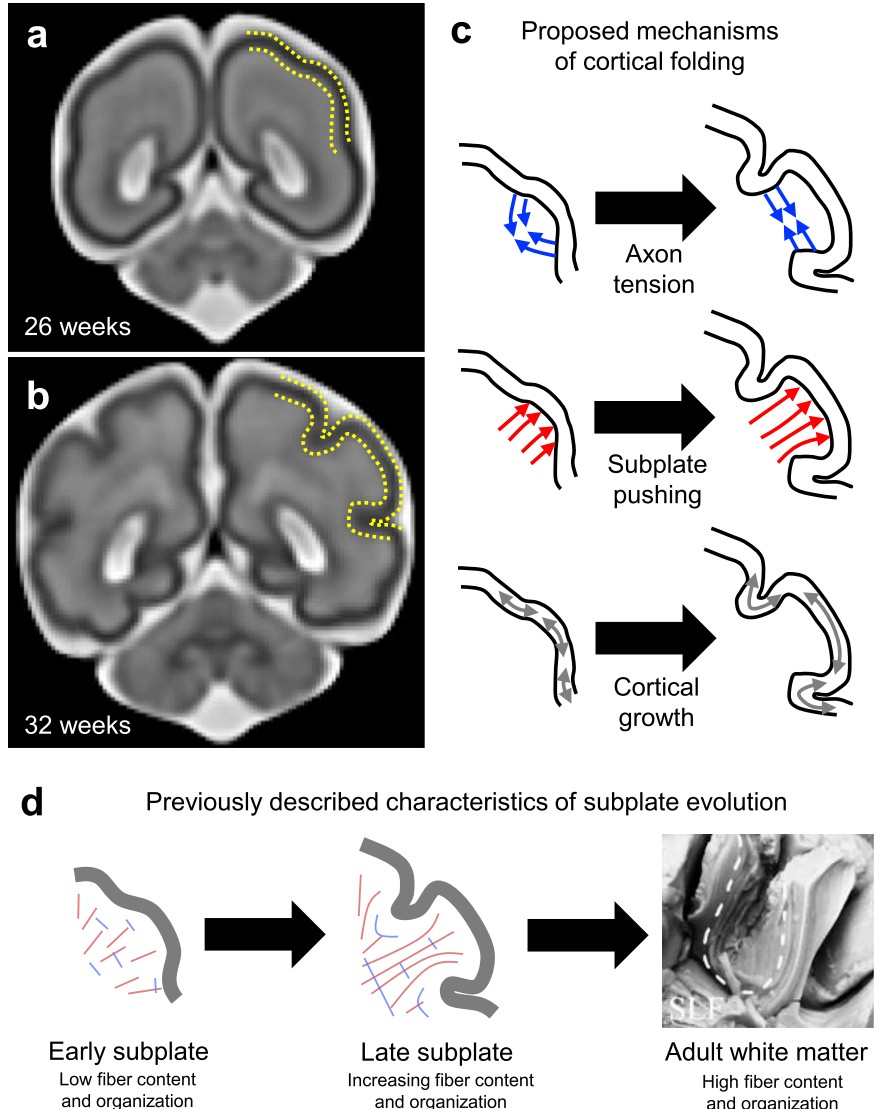

**Fig. 1 Cortical folding and subcortical organization. a–b** Magnetic resonance images of the fetal human brain at the onset of cortical folding (**a**) and after formation of cortical folds (**b**). Images generated from template described in Gholipour et al.[69]. **c** Mechanisms proposed to actively induce cortical folding include axon tension that preferentially tethers specific areas of cortex (top), subplate growth that exerts an outward push to form gyri (middle), and constrained cortical growth, either uniform or patterned, that induces mechanical buckling (bottom). **d** Schematic illustration of previously reported observations in the developing subplate, including steady increase in axon density during the period of folding and emergence of organization mirroring adult white matter. Image of human brain from Burks et al.[41] illustrates predominant organization observed in the adult white matter, including radial organization beneath gyri and tangential organization beneath sulci. Dotted line indicates short association fibers bridging adjacent gyri (U-fibers), SLF = superior longitudinal fasciculus.

However, physical measurements performed in developing brain tissue have found mechanical stresses inconsistent with both proposed theories[14].

In light of these physical findings, recent studies have focused on a third proposed mechanism of folding: that cortical expansion, constrained by slower growth of the underlying subcortical tissue, leads to mechanical buckling (wrinkling or creasing)[15–17]. Within this general framework, constrained cortical growth may be uniform across the cortex, though differential rates have also been proposed to lead to preferential formation of gyri in specific areas[18–20]. Critically, computational simulations of either case predict tissue stresses consistent with experimental observations[14,21]. Several groups[17,22,23] have extended this idea further by exploring the influence of subcortical tissue behavior on cortical expansion-driven folding. Supported by evidence that early subcortical tissues contain axons, and that axons generally lengthen in response to sustained stretch[24–26], subcortical tissues have been defined to grow in response to sustained stretch or stress, similar to a viscoelastic material. Using closed-form solutions and finite element modeling, Bayly and colleagues[22] demonstrated that lissencephaly or polymicrogyria could be explained by a faster or slower stress-dependent growth in subcortical layers. Expanding on this work, Holland and colleagues[23] illustrated that patterned subplate orientations could bias the location and direction of buckling-induced folds. To date, however, such models have considered only predefined subcortical properties and fiber distributions, assumed to remain constant over the period of folding.

In this study, we incorporate dynamically evolving fiber populations into simulations of brain folding, such that axon elongation influences not only the growth of subcortical tissue but also the relative proportion of fibers in each direction. In a departure from past simulation approaches, this framework facilitates predictions related to the evolution of fiber density and orientation observed during development: as specific fiber populations elongate in response to stress, the dominant fiber orientation begins to reflect local, cumulative forces experienced by the tissue. We hypothesize that cortical expansion and folding-induced stresses will lead to stereotyped subcortical fiber orientations consistent with those observed in gyrencephalic brains. To determine whether model predictions are consistent with patterns in developing brain tissue, we performed high resolution ex vivo diffusion tensor imaging (DTI) on fetal rhesus macaque (*Macaca mulatta*) brains at developmental stages in which gyri and sulci are forming, gestational ages (G)85 and G110. Water diffusion in the subplate proximal to the cortical plate was indeed found to exhibit anisotropy, including spatial and temporal trends in agreement with current simulations. Together, results suggest that mechanical stresses are sufficient to induce realistic fiber orientation distributions. Findings provide further evidence in support of cortical expansion-induced folding, as well as a role of tension-induced axon growth in neuronal development that is of relevance to the interpretation of abnormal folding patterns observed in mature brains.

## Results

### Folding stresses induce patterns in subplate fiber networks.
To characterize the influence of mechanical feedback on fiber organization, constitutive equations describing stress-dependent elongation of embedded fibers were employed within a simple model of cortical expansion-induced folding (see Methods for details). A two-dimensional axisymmetric model was selected, and a spherical initial geometry was defined to approximate the curved shape of the unfolded brain (Fig. 2a). This spherical initial shape avoids folding biases induced by spatial variations in curvature and enables precise control over factors considered to bias

the initial location of folding[15,22,23]. Furthermore, an axisymmetric model restricts fold formation to the plane of interest. While this model cannot replicate the branched folding observed in three-dimensions[16], restriction of folding to concentric rings facilitates precise quantification of stresses in the cross-sectional plane as well as along the length of each fold. Material radial, circumferential, and meridional orientations were defined from the initial spherical coordinate system ($e_1$, $e_2$, $e_3$, respectively), and subcortical growth was restricted to elongation of embedded fibers in these orthogonal directions.

To drive folding, cortical growth in each tangential direction ($e_2$, $e_3$) was defined to grow at a rate of $g_c = 0.02$ days$^{-1}$ (d$^{-1}$), based on total cortical surface area change observed during early rhesus macaque development[27]. This corresponds to an approximately 6-fold increase in cortical surface area over a total simulation time of 50 gestational days. Since increase in cortical thickness represents a relatively small component of total cortical growth, the cortex was not defined to grow in the radial ($e_1$) direction, consistent with past modeling studies[17,22,23]. To break symmetry and induce the first fold, a small region of the cortex was defined to grow 5% faster than adjacent cortex, resembling theories of differential cortical expansion in which local or regional variations in cortical growth control the formation of primary gyri[18–20]. As previously reported by Toro and colleagues[17], local differences in cortical growth rate may serve as a complementary influence in cortical expansion-induced buckling models. Alternative perturbations to induce buckling were explored in Supplementary Fig. 1 and Supplementary Table 1.

Figure 2 depicts the progression of subplate organization over the period of cortical expansion and folding, with nondimensionalized simulation time, $T = g_c t$, ranging from 0 to 1. ($T = 0$ corresponds to the onset of cortical expansion, and $T = 1$ corresponds to the maximum simulation time of 50 gestational days.) Prior to cortical folding, cortical expansion induced low levels of tension throughout the subplate (Fig. 2b). As folding commenced, tension that evolved beneath prospective sulci was primarily tangential, while tension that evolved beneath prospective gyri was primarily radial, consistent with previously reported stresses in the developing brain[14]. In the cortex, constrained expansion also induced tangential compression, resulting in elastic deformation and tension in the radial direction. In Fig. 2, this can be observed as an increase in thickness, with the cortical layer exhibiting an approximately 50% increase in thickness at $T = 0.5$ and beyond. In both ferret and human cortices, an approximately 100% increase has been reported over a similar period[28,29]. Thus, elastic deformation was sufficient to account for approximately half of the cortical thickening observed over this period of development.

To match the low fiber content (axons and other processes such as radial glial scaffolds) and weak radial orientation observed in early subcortical layers of gyrencephalic species[30], radial and tangential fiber volume fractions were initially set at $f_{1,0} = 0.12$ and $f_{2,0} = f_{3,0} = 0.1$, respectively, with remaining non-fiber volume fraction (cell bodies, extracellular matrix) defined by $f_{c,0} = 1 - f_{1,0} - f_{2,0} - f_{3,0}$. However, as shown in Fig. 2c, stress-dependent fiber elongation combined with a general state of tension throughout the subplate led to a steady increase in total fiber volume fraction over time. This trend is consistent with reported increases in subcortical fiber content over the period of cortical folding[8]. Furthermore, folding-induced patterns of stress led to changes in the distribution of fibers, resulting in increased tangential fiber volume fraction beneath sulci and increased radial fiber fraction beneath sulci (Fig. 2d). This stereotyped pattern beneath folds is generally consistent with white matter organization observed in the adult human brain (Fig. 1d)[11,12]. Throughout

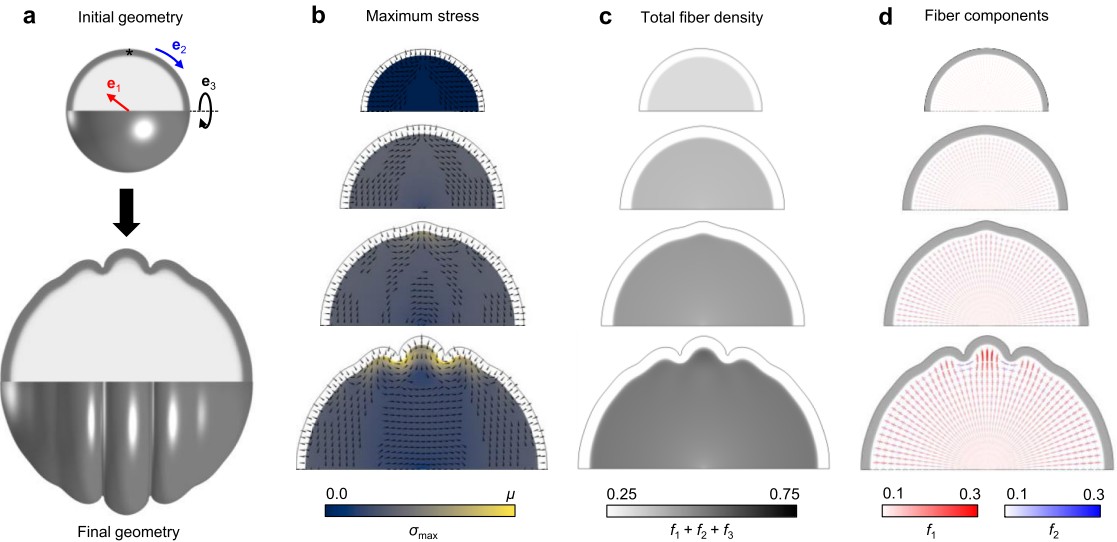

**Fig. 2 Cortical folding induces stereotyped fiber organization beneath gyri and sulci. a** Model geometry before and after folding. For the axisymmetric model, material coordinate directions are defined in radial ($e_1$), circumferential-tangential ($e_2$), and axisymmetric-tangential directions ($e_3$), and can be followed over the course of deformation. The asterisk denotes the location of a symmetry-breaking perturbation. **b–d** Evolution of maximum stress (**b**), total fiber density (**c**), and directional fiber components (**d**) in the subplate over the course of folding. From top to bottom, simulation time $T = 0$, 0.5, 0.75, 1. In **b**, colormap indicates the magnitude of maximum Cauchy stress ($\sigma_{max}$) at each point in the subcortical layer, ranging from 0 to $\mu = 300$ Pa (shear modulus of tissue, as defined in Methods). Tick marks indicate the maximum stress direction, with no visible tick marks indicating the $e_3$ direction. In **c**, greyscale indicates the magnitude of total fiber density, where $f_1$, $f_2$, $f_3$, indicate the fiber volume fraction in directions $e_1$, $e_2$, $e_3$, respectively. In **d**, crosses (+) are used to visualize local fiber volume fraction in terms of radial and (circumferential) tangential directions. Magnitude of fiber volume fraction in these specific directions is represented by color (radial fibers in red, tangential fibers in blue), such that a dark red region indicates increased radial fibers per unit volume, dark blue indicates increased tangential fibers per unit volume, and light color indicates relatively low fiber volume fraction.

the primary analysis presented, non-fiber components were considered to be non-growing. However, growth of non-fiber components was also explored, resulting in similar patterns (Supplementary Fig. 2).

**Initial fiber properties influence cortical fold morphology.** The model in Fig. 2 considers slightly radial initial fiber orientation and low initial fiber density, with a stress-dependent fiber elongation rate of approximately $a = 0.001 \, \text{Pa}^{-1}\text{d}^{-1}$ for all fibers. However, prior neurite elongation experiments, which are diverse in terms of neuron type and magnitude of applied stretch or stress, suggest a wide range of possible rates. Table 1 illustrates the range of previously reported measures as well as a calculated stress-dependent fiber elongation rate, $a$ (see Methods for details). For studies reporting force-dependent elongation, calculated values of $a$ ranged from 0.02 Pa$^{-1}$d$^{-1}$ in chick embryonic forebrain neurons[31,32] to 0.1 Pa$^{-1}$d$^{-1}$ in chick embryonic dorsal root ganglia neurons[33]. For experiments reporting stretch-dependent elongation[23,34], conversion based on measured Young's modulus for individual neurites, ranging 100–4600 Pa[35,36], yields the range $a = 0.0004$–0.02 Pa$^{-1}$d$^{-1}$. Recent experiments have also reported stretch-dependent elongation in ultra-long astrocyte processes designed to mimic radial glial scaffolds, though rates were four to six times slower than those reported in neurites[37].

To probe the influence of this parameter, a nondimensionalized response ratio, $R$, was considered to describe the subcortical fiber elongation rate ($a$) relative to cortical growth rate ($g_c$). $R = 0$ corresponds to no fiber elongation, while larger $R$ denotes faster fiber elongation relative to cortical growth rate. Bayly et al.[22] and Holland et al.[23] explored theoretical effects in the ranges of $R = 1$–40,000 and $R = 10$–300, respectively, under the assumption that this behavior applied to all subcortical tissue.

Figure 3 explores the influence of stress-dependent fiber elongation rate and fiber distributions on the current model, for which elongation behavior is applied to only fiber volume fractions. As illustrated in Fig. 3a, elongation rates on the lower end of the reported range ($a < 0.01$ Pa$^{-1}$d$^{-1}$, $R < 600$) facilitated folding under the current boundary conditions and model parameters. Conversely, when very fast fiber elongation rates were considered, growth of the subcortical tissue kept pace with the expanding cortex and prevented accumulation of the compressive stresses necessary to induce folding. Within the range that induced folding, faster fiber elongation (higher $R$) led to later onset of folding, and slower fiber elongation (lower $R$) led to earlier folding (Fig. 3b). An increase or decrease in the initial fiber volume fraction also resulted in later or earlier onsets of folding, respectively (Fig. 3c). These results are generally consistent with past models incorporating stress-dependent subcortical growth[22,23], which required slower rates to induce folding under similar boundary conditions ($R < 32$) but approximated subcortical tissues to contain the maximum total fiber volume fraction ($f_1 + f_2 + f_3 = 1$).

Notably, fast fiber elongation also appeared to produce smoother, more sinusoidal folds at the onset of folding, while slower elongation rates produced sharper, cusped sulci at the onset of folding (Fig. 3b). Tallinen and colleagues[21] previously showed that sinusoidal buckling, or wrinkling, will only occur in models for which the cortex is stiffer than underlying subcortical layers. By contrast, if cortical stiffness is similar to (or softer than) the underlying tissue, a lower creasing threshold will be reached first, causing an immediate transition to sharp, self-contacting sulcal cusps that deepen over time. While the current model employs similar elastic properties for the cortex and subcortical layers, based on previously reported measures in the developing brain[14,22,38], inclusion of stress-dependent fiber elongation effectively increases compliance of the subcortical tissue. As shown in Fig. 2 ($R = 60$), the initial result is sinusoidal folding (wrinkling), but cusped sulcal morphology is observed at later stages, as folds deepen in the post-buckling regime. This

**Table 1 Parameter range for neurite elongation based on available literature.**

| Cell type | Reported axon caliber | Applied force range | Reported elongation rate (stretch or force-dependent) | | Calculated stress-dependent elongation rate | Model response ratio (dimensionless) |
|---|---|---|---|---|---|---|
| | $D$ (µm) | $F$ (pN) | $b$ (h$^{-1}$) | $G^{axon}$ (µm pN$^{-1}$ h$^{-1}$) | $a$ (Pa$^{-1}$d$^{-1}$) | $R$ (1) |
| Embryonic rat dorsal root ganglia[34] | 0.9 | | 0.076* | | 0.0004–0.02 | 91 |
| Embryonic chick sensory neurons[23] | 1.0 | | 0.08 | | 0.0004–0.02 | 96 |
| Embryonic chick forebrain neurons[31,32] | 1.0 | 10–1000 | | 0.1 | 0.02 | 850 |
| Embryonic chick dorsal root ganglia[33] | 2.0 | 250–5600 | | 0.15 | 0.1 | 5100 |
| P0–P1 murine hippocampal neurons[47] | 5.0 | <10 | | 0.66 | 2.1 | 94,500 |

*Calculated from exponential fit of reported stretch-to-failure experiments.

progression from shallow, relatively sinusoidal undulations to deep, sharp cusps within sulci is consistent with brain morphology over the course of development (Fig. 1a). Thus, current data supports an initial wrinkling instability, followed by deepening and sharpening of sulci into cusps over development.

Related to these effects, we note that initial fiber orientation also played a minor role in the observed morphology of folds (Fig. 3d). When considering the same initial fiber volume fraction and fiber elongation rate, stronger radial organization produced sharper, deeper folds at $T = 1$, and stronger tangential organization produced smoother, more shallow folds. In all cases, the stereotyped fiber organization with respect to folding—more radial beneath gyri and more tangential beneath sulci—was preserved.

**DTI data support stress-induced organization in the subplate.** To validate the stereotyped subplate anisotropy predicted in simulations, high resolution DTI measurements were performed on ex vivo fetal rhesus macaque brains. Primary eigenvectors of the diffusion tensor are shown for data collected from a gestational day (G)85 and G110, of a 168 day gestational term for rhesus macaques (Fig. 4a, b). In these results, the primary eigenvector is expected to be colinear with the dominant fiber orientation within a given image voxel[39].

Figure 4a shows a lateral view of the G85 cerebral cortical surface and a fractional anisotropy (FA) parameter map of a slice oriented as indicated with the yellow dashed line, with primary eigenvector maps within the area indicated by the red rectangle of the inset image. This gestational age is when folding of the cerebral cortex is first initiated, and only rudimentary grooves that will mature into the superior temporal sulcus and sylvian fissure are observable. FA within the cortical plate at this gestational age was uniformly high (approximately 0.7–0.8), similar to previous reports[27,30]. By contrast, FA within the subplate is extremely low (0.1–0.2)[30]. Within the subplate, the primary eigenvector orientation is radial, which may be attributed to the influence of radial glial fibers. No evidence of tangential subplate organization was observed beneath early-stage sulci, contradicting prior interpretations that subplate orientation initiates formation of folds[9,23]. However, this observation is consistent with our model (Fig. 4c) in which the primary eigenvector direction, calculated from fiber volume fractions, remained slightly radial from model initiation (initial fiber volume fractions set such that $f_{1,0} > f_{2,0} = f_{3,0}$) through early folding ($T = 0.75$).

Figure 4b shows the analogous FA map for a rhesus macaque at G110, after primary sulci have formed. Although the magnitude of FA within the subplate remains much lower than within the cortical plate, the orientations of primary eigenvectors within the subplate beneath sulci have changed from radial to tangential, whereas the subplate tissue within gyri remains radial. The tangential orientation of primary eigenvectors within the subplate beneath the sulci is consistent with the tangential orientation found beneath sulci in our model (Fig. 4d). Notably, the magnitude of diffusion anisotropy reflected in FA (Fig. 4b) differs from the magnitude of anisotropy in fiber orientation distributions reflected in FA* (Fig. 4d), with the latter exhibiting a pattern of higher values within gyral subplate. By contrast, diffusion anisotropy in the G110 subplate is uniformly low. There are several possible reasons for this difference, such as a dependence of FA on fiber density and maturation state of myelin[40]. Indeed, at later stages of development, higher FA is observable within developing gyral white matter compared to sulcal regions (e.g., in the G135 brain, as can be observed in Fig. 7b of ref. [40]).

The primary eigenvector maps of Fig. 4 suggest consistency between primary eigenvector orientations measured with diffusion (reflecting fiber orientation) and primary eigenvector orientations calculated directly from simulated fiber volume fractions. However, only a small brain region is shown, representing a projection of 3D information onto a 2D plane. To confirm that the observed orientations are consistent with respect to gyri and sulci throughout the rhesus macaque brain, and that the tangential orientation observed near sulcal fundi are observable irrespective of projection plane orientation, angles between primary diffusion tensor eigenvectors and the local radial directions were calculated throughout the entire brain for four developing subjects. Primary eigenvector angles in the superficial subplate were quantified as shown in Fig. 5, with 0 degrees indicating radial alignment and 90 degrees indicating tangential alignment (parallel to the cortical surface).

For all sulci, primary eigenvectors of the underlying subplate were oriented in a tangential direction. In addition, the subplate beneath crowns of each gyrus consisted of primary eigenvectors with radial orientations. Results indicate that tangential alignment near sulci, and radial alignment near gyri, is a general feature observed within the G110 brain, whereas no tangential alignment was observed prior to formation of sulci in the G85 subplate.

**Deep stresses may influence formation of deep fiber tracts.** Based on the effect of folding-induced stresses on subplate organization, we hypothesized that other mechanical stresses may induce additional features of subcortical fiber organization. To explore this possibility, alternative model geometries were

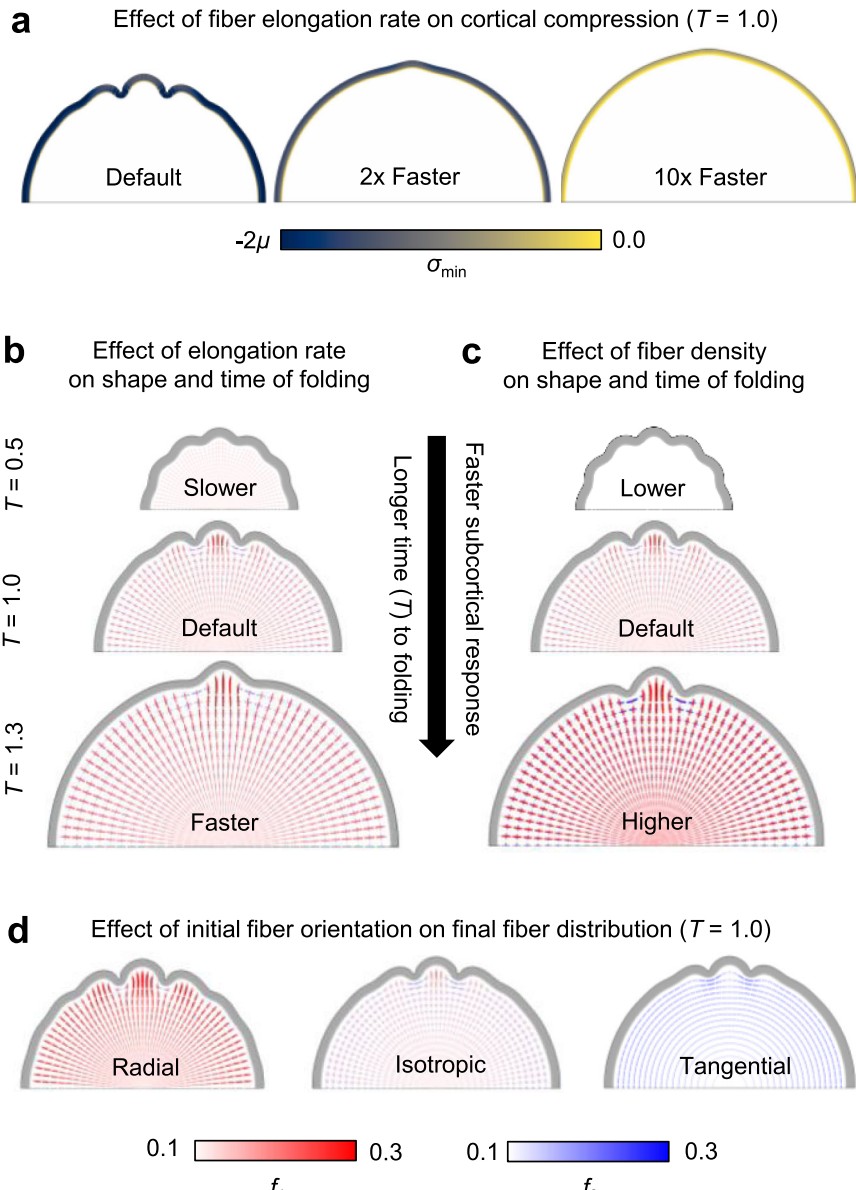

**Fig. 3 Effect of initial fiber properties on cortical folding and final fiber organization. a** Increasing the rate of stress-dependent fiber elongation results in reduced cortical compression and folding at the final simulation time, $T = 1$. From left to right, results are shown for nondimensionalized fiber response ratios $R = 60$ (default), 120, 600. Colormap indicates the magnitude of minimum Cauchy stress ($\sigma_{min}$) at each point in the cortical layer, over a compressive stress range of $-600$ Pa to 0. ($\mu = 300$ Pa represents shear modulus of tissue, as defined in Methods.) No folding is observed in models with $R \geq 600$. **b–c** Rate of fiber elongation and initial fiber density alter the time course and morphology of cortical folding. From top to bottom in **b**, results are shown for nondimensionalized fiber response ratios $R = 30$, 60 (default), 120. From top to bottom in **c**, results are shown for total fiber volume fractions $f_{1,0} + f_{2,0} + f_{3,0} = 0.16$, 0.32 (default), 0.64. For each case shown, the selected simulation time ($T$) was chosen to illustrate morphology and fiber volume fractions at a similar stage of fold maturity. **d** Initial fiber organization produces subtle influence on final fold morphology but similar subcortical fiber organization with respect to gyri and sulci. In **b–d**, crosses ($+$) are used to visualize local fiber volume fraction in terms of radial ($f_1$) and tangential ($f_2$) directions. Magnitude of fiber volume fraction in specific directions is represented by color (radial fibers in red, tangential fibers in blue), such that a dark red region indicates increased radial fibers per unit volume, dark blue indicates increased tangential fibers per unit volume, and light color indicates relatively low fiber volume fraction.

considered to more closely approximate structural features of the developing brain.

To approximate the elongated structure of the temporal lobe or an entire hemisphere (anterior–posterior axis longer than superior–inferior or medial–lateral axes), an ellipsoidal initial geometry was selected. To examine the effect of deep brain structures, two cases were considered: (i) an ellipsoid containing solid tissue, to approximate deep gray matter structures, and (ii) an ellipsoid containing an inner lumen, to approximate a cerebrospinal fluid-filled lateral ventricle. Figure 6 depicts the resulting subcortical organization for each case, considering varying degrees of prolate ellipsoid elongation. No localized increase in cortical growth was applied for these cases, as variations in curvature along the ellipsoidal geometry provide an alternative perturbation to initiate folding. While minimal folding was observed prior to $T = 1$, folding and stereotyped subplate organization became apparent at later time points ($T = 1.25$, Supplemental Fig. 1).

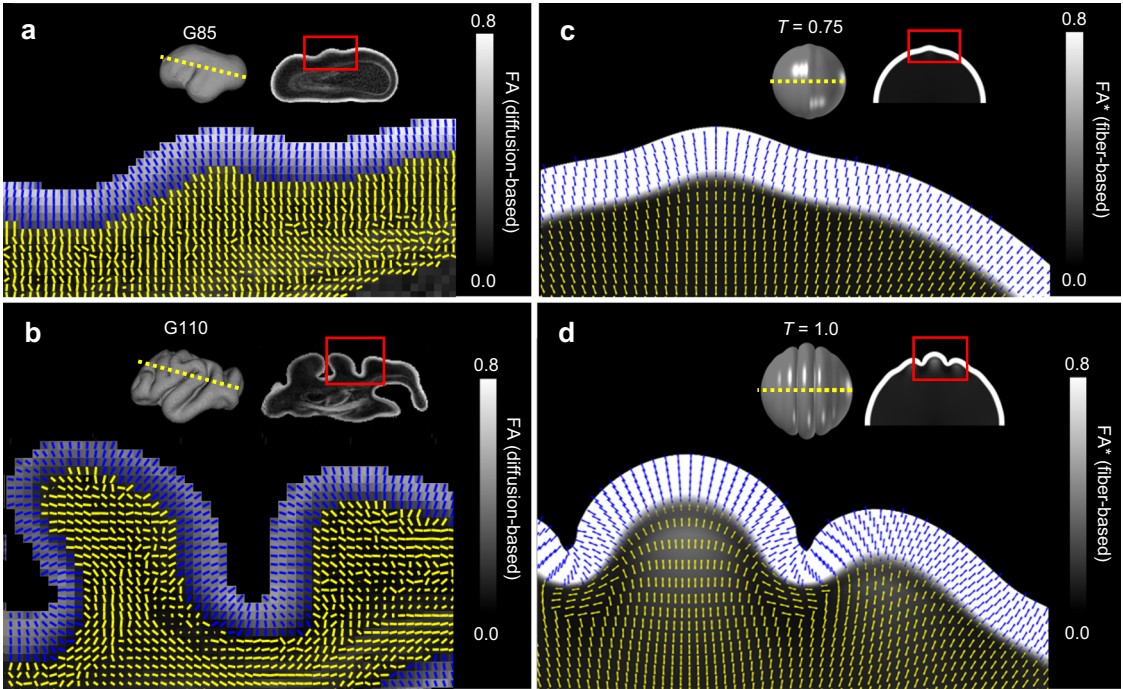

**Fig. 4 Anisotropy in the developing subplate matches model predictions. a–b** Primary eigenvector directions (tick marks) based on high-resolution DTI of the rhesus macaque (*Macaca mulatta*) brain at G85 (**a**) and G110 (**b**), at the slice indicated by the yellow dotted line and location indicated by the red box. Tick marks that appear as dots, rather than lines, indicate primary eigenvector direction that is out of plane (pointing into the page). Fractional anisotropy (FA) maps are shown as an underlay. **c–d** Primary eigenvector directions (tick marks) based on the fiber volume fractions predicted by model at the onset of folding (**c**) and after fold formation (**d**). FA* maps based on the 3D tensor describing model subplate volume fractions are shown as an underlay, with cortical FA* (not defined in model) set to 1 for visualization. Tick marks are colored such that blue = cortex and yellow = subplate and deeper subcortical layers. *T* = nondimensionalized simulation time. Data are presented for the only G85 hemisphere, and for one representative hemisphere of three G110 hemispheres investigated in this study.

As shown in Fig. 6a, cortical expansion generated increased tension in the direction parallel to the long axis of ellipsoid, inducing tangential fiber elongation along the middle, low curvature region of elongated ellipsoidal models. This result is reminiscent of longitudinal fasciculi and optic radiations that span the long axis of the brain in humans[12,41]. In models with an inner ventricle, cortical expansion also induced strong tangential tension near the ventricular surface, leading to a primarily tangential orientation along the full circumference of the ventricle. These trends are consistent with high tensile forces near the ventricle in the developing ferret brain[14] as well as anisotropy previously reported in the rhesus macaque prior to folding[30].

In terms of development over time (Fig. 6b), tangential organization was first apparent along the middle, low-curvature third of the ventricle for elongated ellipsoid models but eventually propagated to the poles as well. To confirm these predictions as they relate to brain geometry, model fiber orientations were again compared to high resolution DTI in the rhesus macaque. As shown in Fig. 7a (blue inset), increased FA and tangentially oriented primary eigenvectors (running parallel to the long axis of the ventricle) were visible near the ventricle in the G85 macaque brain, consistent with previous reports[30]. Furthermore, as quantified in Fig. 7b–d, tangential organization along the ventricle was most apparent in the low curvature portion of the ventricle (along its length), suggesting that organization near the poles may occur later, consistent with the predicted time course in Fig. 6b.

By contrast, models with a solid inner core generally maintained radial organization in deeper layers, with tangential organization only appearing in a mid-depth location for

ellipsoidal models. Radial organization could relate to projection fibers that extend from deeper structures into the cortex, and similar organization was observed in select brain regions (Fig. 7a, red inset). However, this prediction could not be robustly confirmed with our current dataset, as diffusion anisotropy in white matter near striatal nuclei is dominated by early-developing fiber tracts such as the internal capsule and corpus callosum. Presumably, additional biological mechanisms such as molecular axon guidance cues influence the organization of these white matter structures independently of mechanical forces.

## Discussion

Recent experimental and computational advances have increased recognition for the role of mechanics, alongside genetic and neural activity-dependent factors, in development of the cerebral cortex[17,42,43]. However, previous theories and investigations have not fully reconciled fiber organization of the prospective white matter with observed patterns of tension in the developing brain[10,21,23,44–46]. In this work, we characterized the effects of stress-induced axon elongation on the organization of developing white matter fibers during cortical folding. Comparing simulation results to high resolution imaging data, we found that the cortical expansion-driven model of folding is sufficient to predict not only cortical morphology and subcortical stresses, but also diffusion anisotropy patterns observed in the prospective white matter during the period of brain folding.

To accomplish the goals of this study, we proposed a simulation approach to explicitly define and track growing fibers as a distinct tissue component within the subplate. This approach offers several advantages over previous approximations. First,

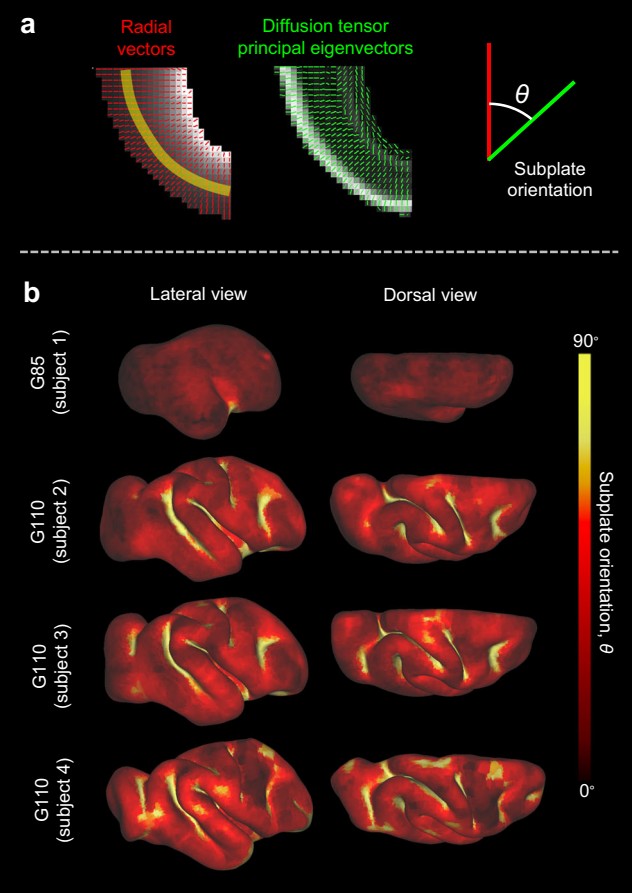

**Fig. 5 Primary anisotropy orientation across subplate in developing rhesus macaque (*Macaca mulatta*). a** Radial orientation vectors generated from a gradient representing distance from the cortical surface (red), and primary eigenvectors generated from DTI (green). For voxels at a set distance from the cortical surface (the superficial subplate, indicated by yellow strip), subplate orientation, $\theta$, was calculated as the angle between the radial vector (red) and diffusion tensor principal eigenvector (green). **b** Subplate orientation, denoted by color, mapped onto cortical surface reconstructions for one G85 (top) and three individual G110 (bottom) brains. Panel **a** modified from Wang et al.[30]. Data are presented for the only G85 hemisphere, and for all 3 G110 hemispheres investigated in this study.

fiber representation enables simulation of tissue remodeling, with the goal of recapitulating the dramatic evolution of subcortical fibers observed during the period of folding[8–10]. As illustrated in both the finite-element simulations of cortical folding (Figs. 3, 4, and 6) and simpler theoretical cases (Fig. 8 in Methods), sustained subplate tension predicts increasing fiber content consistent with observations during development[8]. Second, this approach enables more direct application of fiber elongation behavior reported in previous experimental work. Review of previous elongation experiments revealed considerable variability in reported stretch- or tension-dependent axon elongation rates (Table 1). Even among methodologically similar studies, a wide range of elongation rates have been observed for different neuron types at different stages of maturation[31–33,47]. These differences may relate to availability of biological resources such as cytoskeletal proteins required to elongate neurites, which may be developmentally regulated[48]. Still, previous models have relied on stress-dependent growth rates below the range of reported measures to induce folding under conditions representative of the brain[22,23].

By applying axon elongation rates only to fiber components, models in the present study confirm that axon elongation rates in the range of previously reported measures can induce basic folding morphologies (Fig. 3). Notably, however, only rates at the lower end of the reported range produced a folded cortex in our current model. This result may suggest a slower response of neurites under in vivo conditions during the period of folding, with in vitro experimental conditions (including relatively rapid stretches compared to those anticipated during brain development) representing the upper bound of axon elongation behavior. Moreover, simulations presented here do not distinguish fiber contributions from neuronal and non-neuronal sources (neurites versus radial glial scaffolds). Radial glial processes have been observed to extend at rates comparable to axonal growth cones[49], and astrocyte processes designed to mimic radial glial processes have recently been shown to elongate under sustained tension in vitro, although at a slower rate[37]. The need for values on the lower end of the experimentally reported range may therefore reflect an early predominance of slower-growing fibers such as radial glial scaffolds[37]. Additional studies are needed to clarify the rates of tension-induced elongation of various fiber types in the context of brain folding, as well as the precise volume fractions of these components during development, in relevant gyrencephalic model systems such as the developing ferret or rhesus macaque.

Consideration of fiber elongation rates and fiber volume fractions may improve our understanding of the relationships between abnormal brain morphologies and abnormal white matter organization. Previously, Bayly and colleagues[22] showed that faster subplate response relative to cortical growth can lead to decreased folding (lissencephaly or pachygyria), while slower subplate response can lead to many small folds (polymicrogyria). However, both lissencephaly and polymicrogyria have also been associated with reduced short-range association fibers in human[29,50]. In the current framework, slower fiber elongation (or faster cortical growth) predicts not only increased gyrification but also reduced overall fiber density (Fig. 3b). This is generally consistent with reduced short and long-range connections reported in polymicrogyria[50]. Meanwhile, faster fiber elongation (or slower cortical growth) predicts reduced gyrification and high overall fiber density, but a reduction in tangentially oriented fibers beneath sulci as a consequence of reduced gyrification. This is generally consistent with the reduction of short-range cortico-cortical pathways and preservation of radially oriented fibers (projection pathways) observed in lissencephaly[29]. Finally, our simulations suggest that a reduction in initial fiber density will also produce increased gyrification (Fig. 3c). This is consistent with experimental studies in which early disruption of subcortical connectivity (reduction of intact fibers) led to an overproduction of cortical folds[5,51,52]. Alternatively, as suggested by previous work[17], disruption of connectivity may alter a complex array of developmental mechanisms, such as cortical growth, which could also alter the observed pattern of folding.

It is important to acknowledge that the current model only approximates growth by focusing on specific developmental mechanisms, such as tension-induced fiber elongation and cortical expansion. For example, it does not account for axon growth resulting from other biological mechanisms such as well-established molecular axon guidance factors[53]. Indeed, prior to cortical folding, afferent axons from cortical and subcortical structures form synapses within the subplate, and efferent axons project from the cortex based on non-mechanical cues. The present model's failure to reproduce the complex organization observed near solid deep brain structures, such as the striatum, highlights the need for additional organizing factors to fully understand brain development. Furthermore, the role of

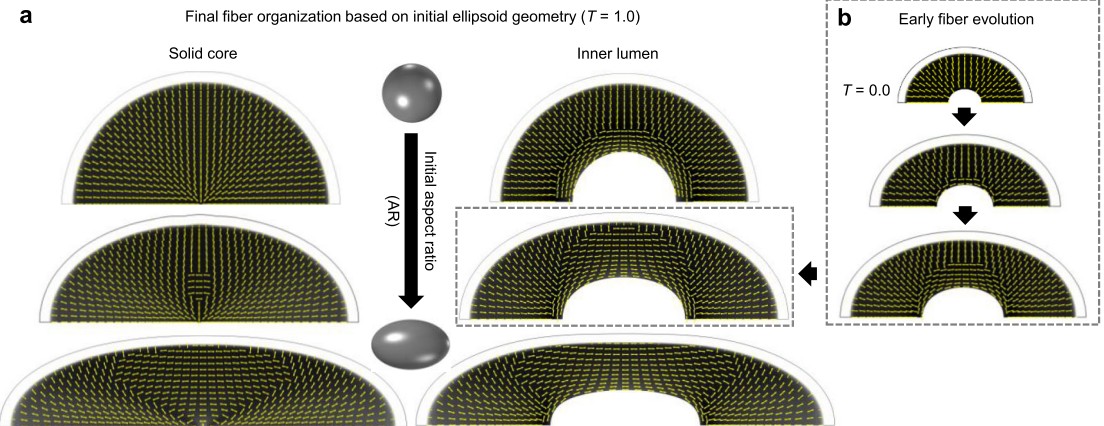

**Fig. 6 Geometric factors induce reorganization of deeper subcortical layers. a** Deep fiber organization resulting for ellipsoidal cases of increasing elongation, as reflected by initial geometry aspect ratio (AR). From top to bottom, initial $AR = 1.01, 1.10, 1.25$. Results are shown for solid ellipsoids (left) and ellipsoids containing an inner lumen (right) at simulation time $T = 1$. For prolate ellipsoidal geometries, cortical expansion induces tension parallel to the long axis and reorganization of fibers in this direction. For models containing an internal lumen as opposed to solid core, tangential tension also develops at the interface, with reorganization of fibers to run along the ventricular surface. **b** Evolution of deep fiber organization in an ellipsoidal model with internal lumen, approximating the shape of the brain and lateral ventricle. From top to bottom, simulation time $T = 0, 0.5, 0.75$.

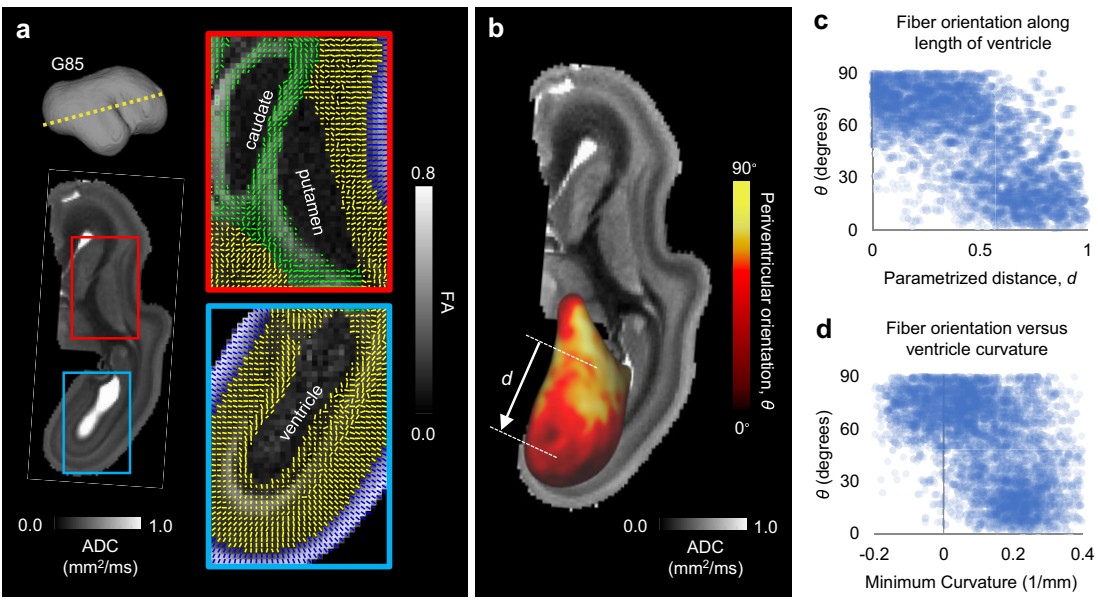

**Fig. 7 Primary anisotropy orientation near the ventricular surface in the G85 rhesus macaque brain. a** Primary eigenvector directions (tick marks) based on high-resolution DTI of the rhesus macaque (*Macaca mulatta*) brain at G85, at the slice indicated by the yellow dotted line. An apparent diffusion coefficient (ADC) map is shown to visualize deep gray matter structures. Red and blue insets illustrate primary eigenvector directions surrounding solid internal brain structures (red) and the fluid-filled lateral ventricle (blue), with fractional anisotropy (FA) maps as an underlay. **b** The periventricular orientation angle, $\theta$, was calculated and projected to the ventricular surface following the same process described in Fig. 5a. ADC map is again shown to visualize surrounding structures. **c** For each vertex on the ventricular surface, parameterized distance ($d$) from the approximate center of the long axis of the ventricle was calculated, ranging from 0 to 1. Scatter plot illustrates the relationship between parameterized distance from center, $d$, and $\theta$. **d** Minimum curvatures were also calculated at each point, ranging from approximately zero near the center of the long axis of the ventricle to approximately 0.2 mm$^{-1}$ near the end of the ventricle. Scatter plot illustrates the relationship between minimum curvature and $\theta$. Data are presented for the only G85 hemisphere investigated in this study. Data points in panels **c–d** represent the values at individual vertices along the ventricular surface in panel **b** (5814 unique vertices).

mechanical feedback may extend beyond axon elongation. Its influence on biological processes such as neurite branching and cell proliferation[43,46] provides increasing support for the idea that mechanical factors act in concert with other factors (eg., innate genetic determinants) during cortical morphogenesis[42,54]. These factors will be important to consider in future models of cortical and subcortical development.

Despite these limitations, the present study provides several lines of evidence to support a cortical expansion-driven, mechanobiological explanation for specific aspects of subcortical organization during development. This framework, in which folding-induced stresses influence the organization of axonal fibers, lies in contrast to alternative hypotheses that axonal organization drives folding (Fig. 1c).

**a**

Theory for tension-induced fiber organization

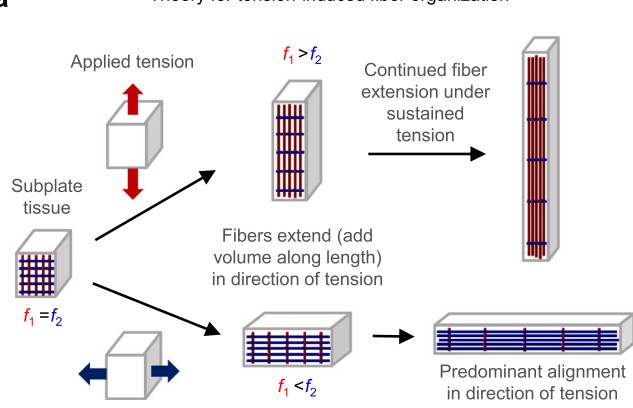

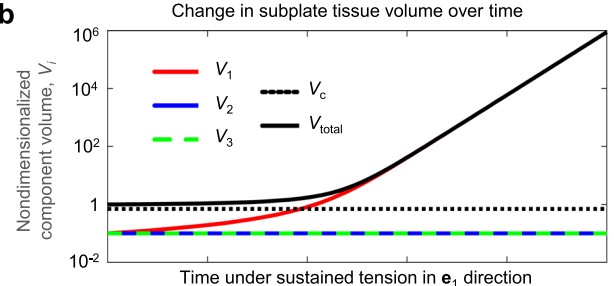

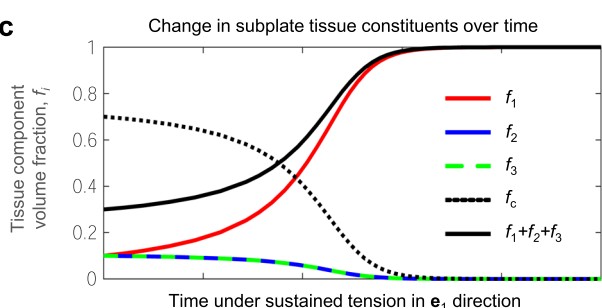

**Fig. 8 Theoretical growth and remodeling in response to sustained tension. a** Schematic illustration of tissue elements with embedded fibers in initial configuration (left), after a short period of stretch and fiber elongation (middle), and after an extended period of stretch and fiber elongation (right). Tissue blocks are not shown to scale since volume increase becomes dramatic over an extended period of stretch. Theoretical behavior of a tissue element subjected to sustained tension in direction $\mathbf{e}_1$ as time approaches infinity is described in **b–c**. **b** Volume associated with each tissue component as a function of time, starting from a nondimensionalized total tissue volume ($V_{total}$) of 1. $V_1$ represents volume associated with fibers aligned in the direction of tension ($\mathbf{e}_1$), while $V_2$ and $V_3$ represent the volume associated with fibers perpendicular to tension ($\mathbf{e}_2$ and $\mathbf{e}_3$), and $V_c$ represents the volume associated with cells and extracellular matrix. Vertical axis is log scaled to facilitate visualization, as total volume begins to increase dramatically (approximately exponentially) at later time points. **c** Volume fraction of each tissue component as a function of time, starting from an initial fiber volume fraction of 10% in each direction ($f_1 = f_2 = f_3 = 0.1$). When tension and fiber growth are allowed to continue indefinitely, fiber volume fraction approaches 1 while the volume fraction of non-fiber components ($f_c$) approaches 0.

First, the pattern of subplate tension predicted by cortical expansion-driven models (greatest along tangential directions beneath sulci and radial directions beneath gyri) is consistent with the directions of tension previously observed in the developing ferret brain[14]. By contrast, competing hypotheses of subcortical

organization-driven folding predict stress patterns inconsistent with tension measurements[43]. For example, if axon-generated tension pulled two regions of cortical plate together to form the walls of a gyrus or tether sulci[6], tension would be higher tangentially below gyri and radially below sulci. Alternately, if the subplate exerted outward force on regions of the cortical plate destined to become gyral crowns[8–10], radial compression would be expected below gyri. In both cases, predicted stresses are inconsistent with the patterns of tension observed experimentally and in the present model.

Second, high resolution DTI data from ex vivo fetal rhesus brain tissue exhibits a consistent pattern across all gyri and sulci at the gestational age in which cortical folds have recently formed (G110, Figs. 4 and 5). Within subplate tissue, the primary eigenvector of the diffusion tensor is parallel to the tangent plane of the pial surface beneath sulci, while primary eigenvectors are radially oriented in subplate beneath gyral crowns. This is consistent with model predictions of more abundant (higher volume fraction) tangentially oriented fibers at the base of sulci and more abundant radially oriented fibers in the crowns of gyri. Observations are also consistent with tractography studies using fetal human DTI[9] that have reported fibers sweeping tangentially beneath sulci and radially into gyral crowns, similar to U-fibers observed in adult white matter (Fig. 1d). By contrast, the hypothesis that axons pull the cortex to form gyri would predict a reversed pattern of predominant axon orientations[6,13,46].

Third, contrary to interpretations suggesting subplate heterogeneity precedes and influences cortical folds[8–10], we found no evidence of tangential subplate orientation prior to folding, even once emerging sulci could be clearly delineated (G85, Figs. 4 and 5). Previous simulations of cortical expansion-induced folding have illustrated the potential for pre-existing subplate orientations to bias the location and direction of folds[23], and this general behavior can be reproduced with the present model (Supplementary Fig. 2 and Supplementary Table 1). However, such patterns are not observable at the onset of folding, nor are they necessary to explain the observed link between folding and subplate organization. Fiber remodeling induced by folding stresses may even be sufficient to override weak heterogeneities in subplate orientation (Supplementary Fig. 2).

Finally, we explored the possibility that mechanical stresses other than those induced by folding may contribute to white matter organization. In ellipsoidal models, we observed increased tangential tension (parallel to the long axis) prior to the onset of folding. The resulting accumulation of longitudinal fibers was strikingly similar to deep longitudinal fasiculi that span the long axis of the human brain: the superior longitudinal fasiculus, which runs from the frontal to occipital lobe, and the inferior longitudinal fasiculus, which follows the length of the temporal lobe[11,12]. Furthermore, models with an internal lumen predicted tangential fiber accumulation near the ventricle, with earlier accumulation along the middle of the ellipsoidal structure and later accumulation at the poles. DTI of the G85 macaque confirmed similar predominance of tangentially oriented tissue components in the intermediate zone between the occipital and parietal lobes (Fig. 7)[30], and eventual organization along the full perimeter of the ventricle has been reported in later stages of development[55,56]. This organization, which occurred prior to folding, is consistent with well-established axon tracts that wrap around the lateral ventricles in both gyrencephalic and lissencephalic species.

Together, results of this study support a framework by which mechanobiological cues, resulting from expansion and folding of the cortex, contribute to growth and organization of prospective white matter tissue.

## Methods

**A constitutive model for stress-induced fiber elongation**. To model subcortical reorganization based on mechanical feedback, constitutive equations were

developed to describe the stress-dependent elongation behavior of fibers (axons and other processes such as radial glial scaffolds) embedded in a soft tissue (cell bodies and extracellular matrix).

In continuum mechanics, observable deformation of biological tissue can be conceptualized as a combination of growth, the deformation due to increases in size or number of cells and cell processes, and elastic deformation, the deformation due to mechanical tension or compression[57]. The total observable deformation is described mathematically by the three-dimensional (3D) tensor

$$\mathbf{F} = \mathbf{F}^* \cdot \mathbf{G} \tag{1}$$

where $\mathbf{F}^*$ is the elastic deformation tensor and $\mathbf{G}$ is the growth tensor. In this study, the 3D growth tensor, $\mathbf{G}$, is defined to include orthogonal growth components such that

$$G = G_1 \mathbf{e}_1 \mathbf{e}_1 + G_2 \mathbf{e}_2 \mathbf{e}_2 + G_3 \mathbf{e}_3 \mathbf{e}_3. \tag{2}$$

In subcortical layers, growth is defined in terms of increase or decrease in length of embedded fibers, which occurs in response to mechanical tension or compression along each fiber's axis. This response has been studied extensively in axons from a variety of adult and embryonic sources (Table 1), and more recently in fibers of non-neuronal cells such as astrocytes[37]. Fibers oriented in a given direction are defined to represent a fraction of the total subcortical tissue volume. Thus, at each time point, overall tissue growth in each orthogonal direction ($G_i$) is proportional to not only the applied tension, but also the volume fraction of fibers oriented in the direction of applied tension. The fiber volume fraction in each orthogonal direction ($f_i$) provides a composite measure reflecting the number, length, and diameter of axons.

Based on these criteria, growth in each orthogonal direction ($i$) can be defined by the instantaneous growth rate

$$\frac{\partial G_i}{\partial t} = f_i a (\sigma_i - \sigma_0) G_i \tag{3}$$

where $\sigma_i$ is the mechanical stress experienced by the tissue in direction $i$, $\sigma_0$ is the target stress, and $a > 0$ is stress-dependent fiber elongation rate. More complex models could be developed to account for fiber scatter and the resulting shear growth contributions. However, such components are expected to have minor impact and have thus been omitted from previous simulations of stress-dependent subplate growth[22]. For the simulations in this study, target stress (the threshold above which stress-dependent growth occurs) was set at zero, consistent with experimental studies suggesting an extremely low threshold for tension-based neurite elongation in embryonic forebrain neurons[31,32,47] as well as previous models[14,22]. Mechanical stress, $\sigma_i$, can be defined from the elastic deformation tensor, $\mathbf{F}^*$, as described in the next section.

According to Eq. 3, a block of tissue containing 100% fibers in direction $i$ ($f_i = 1$) will experience maximal growth for a given elongation rate and stress. Conversely, a block of tissue containing 0% fibers will experience no growth regardless of the applied stress. In these unique cases, fiber volume fraction will remain the same over time. However, in cases where multiple tissue components exist and grow at different rates, volume fractions change over time. To account for fiber remodeling, volume fractions must be updated at each time point to reflect the current volume of each component divided by the current volume of the overall tissue. Mathematically, we consider nondimensionalized ratios of current volume relative to the initial volume, with $V_{\text{total}}$ representing the current total tissue volume divided by the initial total tissue volume ($V_{\text{total}} = 1$ at $t = 0$) and $V_i$ representing the current volume of fibers oriented along direction $i$ divided by the initial total tissue volume ($V_i = f_i$ at $t = 0$).

According to these relationships, fiber volume fractions evolve over time according to the function

$$f_i(t) = \frac{V_i}{V_{\text{total}}} = \frac{G_i(t) f_{i,0}}{\sum_{j=1}^{3} G_j(t) f_{j,0} + f_{c,0}} \tag{4}$$

where $f_{i,0}$ is the initial volume fraction of fibers in direction $i$, and $f_{c,0}$ is the initial volume fraction of non-fiber components. For simplicity, the initial volume associated with cells and extracellular matrix is assumed to be non-growing ($V_c = $ constant) throughout the main analysis in this study.

The theoretical consequences of these governing equations are illustrated in Fig. 8, considering a simple block of tissue subjected to sustained tension in one direction and zero stress in perpendicular directions. To represent relatively sparse axonal fibers in the early subplate[8], fiber volume fractions are set low at the outset ($f_1 = f_2 = f_3 = 0.1$ at $t = 0$). As fiber components parallel to the direction of tension elongate, growth is observed in the direction of tension (Fig. 8a). Since fibers are only defined to elongate, not to increase in cross-section (caliber), no tissue growth is observed in directions perpendicular to the applied tension. As fibers increase in length while maintaining their caliber, the corresponding fiber volume increases (Fig. 8b), and the relative volume fraction of fibers in this direction slowly approaches 1 as time approaches infinity (Fig. 8c).

For any direction of sustained tension, the above equations suggest an increase in total fiber volume fraction approaching 1 as time approaches infinity (Fig. 8c). Notably, however, fiber volume fraction in in the adult white matter is known to be closer to 0.8, with extracellular matrix (ECM) accounting for approximately 15–20% of tissue volume[58,59]. To define a maximum fiber volume fraction <1, growth behavior may be modified to produce a set volume of ECM alongside new fiber volume, as described in Supplementary Information (Supplementary Fig. 2).

## Finite element models of folding

All finite element simulations were performed using COMSOL Multiphysics software (version 5.3a, COMSOL Inc., Burlington, MA). Both spherical and ellipsoidal models utilized an axisymmetric domain and assumed quasi-static, time-dependent solutions. As in previous models of brain folding[14,21], the elastic response of developing brain tissue was modeled as a standard neo-Hookean material using the strain energy density function:

$$W = \frac{\mu}{2} \left( J^{*-\frac{2}{3}} \mathrm{tr}(\mathbf{F}^{*\mathrm{T}} \cdot \mathbf{F}^*) - 3 \right) + \frac{\kappa}{2} (J^* - 1)^2 \tag{5}$$

where $\mathbf{F}^*$ represents the elastic deformation tensor and $J^* = \det(\mathbf{F}^*)$. Here we considered a shear modulus $\mu = 300$ Pa and bulk modulus $\kappa = 100\mu$ (Poisson ratio ~0.49), based on material properties derived from previous studies[14,22,38]. The same elastic properties were applied to all cortical and subcortical tissue components, including embedded fiber volume fractions. Anisotropic elastic properties could be incorporated by choosing an alternate form of $W$ (Eq. 5), and several such forms have been proposed for adult brain tissue[60]. However, no studies to date have confirmed mechanical anisotropy in developing subcortical tissues nor increased stiffness of unmyelinated axons relative to surrounding subplate tissues. In the absence of this data, we assume isotropic elastic behavior, consistent with past models of cortical folding[14,16,21,22].

To simulate cortical surface area expansion, cortical growth was defined to occur at predefined rate, as a consequence of biological processes including the intercalation of cells into the cortex[61] and tangential expansion of cortical neuropil[43], such that

$$\frac{\partial G_1}{\partial t} = 0, \frac{\partial G_2}{\partial t} = g_c G_2(t), \frac{\partial G_3}{\partial t} = g_c G_3(t). \tag{6}$$

These equations correspond to exponential growth in both tangential directions, $G_2(t) = G_3(t) = e^{g_c t}$, and no growth in the radial direction, $G_1(t) = 1$. Cortical growth rate, $g_c$, was estimated from an exponential fit of total surface area change observed during early rhesus macaque development[27]. Previous models have considered both linear[16,22,23] and nonlinear[17] cortical growth. Although much remains unknown from a mechanistic perspective, past studies have noted nonlinear expansion: e.g., a slow linear growth phase associated with cell migration and somal intercalation followed by a fast phase that is coincident with cytoarchitectural differentiation[62].

For simulations in which cortical growth was not uniform, a very small, localized increase in cortical growth ($g_c^*$) was added according to the function $g_c^* = 0.05 \, g_c \, (\cos \Theta)^{1000}$ where $\Theta$ describes the angle from edge to edge of the simulated curved surface (−90 to +90 degrees). As illustrated in Supplementary Fig. 1, the taper of this peak falls well within the bounds of an induced gyrus and did not affect the wavelength of folding. For simplicity, the 5% increase in growth for this region was maintained throughout the simulations.

As described in Eq. 3, growth of subcortical layers was defined by stress-induced fiber elongation, where $\sigma_i$ ($i = 1$ to 3) represent the radial, circumferential, and meridional components of the 3D Cauchy stress tensor, $\boldsymbol{\sigma}$. This stress tensor is defined from the elastic deformation component of the total deformation tensor as

$$\sigma = J^{*-1} \mathbf{F}^* \cdot \frac{\partial W}{\partial \mathbf{F}^{*\mathrm{T}}}, \tag{7}$$

where $J^* = \det(\mathbf{F}^*)$, the ratio of volume change due to elastic deformation.

## Parameter determination for stress-dependent fiber elongation

The range of potential values for the stress-dependent fiber growth parameter, $a$, was calculated based on reported measures from previous studies (Table 1). For studies that reported force-dependent elongation rate ($b$, change in length per unit force per unit time), $a$ was directly calculated as $a = bA/L$, where $L$ represents the reported initial neurite length and $A$ represents the neurite cross-sectional area based on reported axon caliber (diameter), $D$. For studies that reported stretch-dependent elongation rate ($G^{\text{axon}}$), $a$ was approximated by assuming the stress–strain relationship for uniaxial stress and small deformations ($\sigma = E\varepsilon$, where $\varepsilon$ represents strain along the axis and $E$ represents the Young's modulus of fibers) such that $a = G^{\text{axon}}/E$. To account for the wide range of Young's modulus values previously reported for axons[35,36], the range of corresponding values was calculated and reported in Table 1.

To bring past measures and models into a consistent framework, we also defined $R = G^{\text{axon}}/g_c$ as a nondimensionalized parameter representing the subcortical fiber response rate relative to the cortical growth rate. Under the assumption of an incompressible material undergoing small deformations and uniaxial tension, the stress–strain relationship for model fibers was again approximated as $\sigma = E\varepsilon$, with $E = 3\mu$ approximating the Young's modulus of model fibers. Thus, for a given value of $a$, $G^{\text{axon}} = 3a\mu$ and $R = G^{\text{axon}}/g_c = 3a\mu/g_c$. This nondimensionalization approach is similar to the approaches described by Bayly et al.[22] and Holland et al.[23], though ratios of these studies considered a linear cortical growth rate. Models in this study focused on the ratio $R = 60$ (Figs. 2, 4, and 6, Supplemental Figs. 1–3), and behavior across a range of values was explored in Fig. 3. Comparisons relative to linear cortical growth rate are explored in Supplementary Fig. 3.

## Magnetic resonance imaging and analysis

Ex vivo MRI scans were performed on four rhesus monkey brain hemispheres, one from a fetus perfusion-fixed at G85, and three from three other fetuses perfusion-fixed at G110. All brain hemispheres were obtained from control animals described in ref. [63]. All procedures involved in the

generation of brain tissue used for this study were conducted in accordance with the Guide for the Care and Use of Laboratory Animals and the National Institutes of Health Guidelines for the Care and Use of Laboratory Animals. These procedures were approved by the Oregon National Primate Research Center Institutional Animal Care and Use Committee. Postmortem fetal brain hemispheres were immersed in Fluorinert Electronic Liquid FC-77 (3 M, St. Paul, MN) during MRI scans and returned to PBS immediately afterwards. Experiments were performed on an 11.7 T small-animal MRI system (Bruker, Rheinstetten, Germany) and a custom Helmholtz coil (5 cm diameter, 5 cm length) was used for radiofrequency transmission and reception. A multi-slice spin-echo pulse sequence (TR/TE = 15 s/30 ms), incorporating a Stejskal–Tanner diffusion sensitization gradient pair was used to acquire diffusion MRI data at an isotropic resolution of 0.3 mm for G110 hemispheres and 0.2 mm for the G85 hemisphere. A 25-direction, icosahedral sampling scheme[64] was utilized for all experiments with 3 b0 images, and diffusion weighted images with a $b$-value of 2500 s/mm$^2$. Standard procedures using the Bruker Paravision software (version 5.1) were followed to calculate eigenvalues ($\lambda_1$, $\lambda_2$, and $\lambda_3$, listed from smallest to largest) and eigenvectors ($v_1$, $v_2$, and $v_3$). DTI indices such as FA and apparent diffusion coefficient (ADC) were calculated from the eigenvalues for each voxel.

Brain segmentation was achieved by first intensity thresholding and then manual editing on the ADC maps using ITK-SNAP (http://www.itksnap.org[65]). The resulting segmentations were used as input for the SureFit operation in the CARET software package (http://brainvis.wustl.edu[66]) to generate cortical pial surface representations.

To determine the dominant fiber orientation relative to the local radial direction ($e_1$) of a given voxel, $\theta$, the angle between $v_1$ and the cortical surface norm, were estimated as illustrated in Fig. 5a. First, a distance matrix from pial surface was computed for each voxel from the whole brain segmentation. Then, a 3D gradient of the distance matrix was derived and, for a given voxel, oriented along the radial direction. Angle $\theta$, with the unit of degree and range of 0–90, was calculated for each voxel. A small $\theta$ indicates $v_1$ is more radially aligned whereas a large $\theta$ indicated $v_1$ is more tangentially aligned relative to cortical surface. To visualize $\theta$ within the superficial subplate across gyri and sulci, the superficial subplate (layer 4 in ref. [30]) was manually segmented and $\theta$ within this region were averaged from a radius of 2 voxels and projected to the nearest cortical nodes.

**Primary fiber orientation and fractional anisotropy.** For MRI data obtained in the rhesus macaque, primary eigenvectors and fractional anisotropy (FA) were calculated using standard definitions[67]. While these diffusion-based values offer an indirect measure of tissue properties, they are thought to generally reflect primary fiber orientations and tissue anisotropy resulting primarily from fiber distributions, though other factors such as myelination, volume fractions of various tissue components, and membrane permeability, also likely influence these maps. Thus, as a first approximation for simulation comparisons, primary eigenvectors and fractional anisotropy were calculated from fiber and non-fiber tissue volume fractions.

First, a matrix describing the initial (orthogonal) fiber volume fractions was defined in the reference coordinate frame as:

$$\mathbf{A} = \begin{bmatrix} f_1 & 0 & 0 \\ 0 & f_2 & 0 \\ 0 & 0 & f_3 \end{bmatrix} \tag{8}$$

The coordinate transformation $\mathbf{A'} = \mathbf{F}^{\mathrm{T}} \cdot \mathbf{A} \cdot \mathbf{F}$ was applied account for rotations from the reference (initial) to deformed (final) configuration, which induces shear (non-orthogonal) components in the deformed configuration. Principal eigenvectors and eigenvalues of $\mathbf{A'}$ ($v_i'$ and $\lambda_i'$, respectively) were calculated using COMSOL built-in functions. Similarly, the deformed radial direction was tracked with the coordinate transformation $\mathbf{u}_1' = \mathbf{F} \cdot \mathbf{u}_1$ to allow calculation of angle $\theta$ between the radial direction, $\mathbf{u}_1'$, and first principal eigenvector, $v_1'$.

Fractional anisotropy, which is 0 for a fully isotropic tissue and 1 for a maximally anisotropic tissue, incorporates information from not only fiber volume fractions, but also the isotropic tissue components represented by $f_c$, which is distributed equally among the three principal eigenvector directions. Thus, simulated fractional anisotropy, FA$^*$, was defined at each point as follows

$$\mathrm{FA}^* = \sqrt{\frac{(\lambda_1^* - \lambda_2^*)^2 + (\lambda_2^* - \lambda_3^*)^2 + (\lambda_1^* - \lambda_3^*)^2}{2(\lambda_1^{*2} + \lambda_3^{*2} + \lambda_3^{*2})}} \tag{9}$$

where $\lambda_i^* = \lambda_i' + f_c/3$, to account for the volume fraction of isotropic tissue that is distributed equally among the three principal eigenvector directions.

**Reporting summary**. Further information on research design is available in the Nature Research Reporting Summary linked to this article.

## Data availability
The data that support these findings, including model files and processed MRI data generated in this study, have been deposited in Zenodo[68] (https://doi.org/10.5281/zenodo.5573193). Source data for plots in Fig. 7c, d are provided with this paper.

## Code availability
All finite element simulations were performed using COMSOL Multiphysics commercial software (version 5.3a, COMSOL Inc., Burlington, MA). Model files incorporating user-defined behaviors, as well as custom MATLAB codes used in MRI data analysis and visualization, have been made available as part of the Zenodo dataset[68] (https://doi.org/10.5281/zenodo.5573193).

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

## Acknowledgements
This work was supported by the National Institutes of Health R01 award NS111948 (K.G, C.K., X.W.), R01 AA021981 (C.K.), and the National Science Foundation award DMS-2011274 (C.K.). We thank Philip Bayly, Victor Borrell, and Anthony (Paul) Barnes for insightful conversations related to this work.

## Author contributions
K.G. conceived the hypothesis and developed the mathematical theory and computational models. C.K and X.W. conceived and designed experimental measures for rhesus macaque. X.W. performed experiments and measurement of DTI data. All authors contributed to interpretation of results. All authors read and edited the manuscript.

## Competing interests
The authors declare no competing interests.
