## [Peer Review File · Nature Communications]

A model of tension-induced fiber growth predicts white matter organization during brain foldingREVIEWER COMMENTS

Reviewer #1 (Remarks to the Author):

I think this is a well written manuscript describing an interesting set of models and ex vivo observations that, together, make a substantial contribution to the literature on cortical folding. The authors have presented evidence for a novel idea, namely that mechanical forces created by cortical folding (buckling forces) affect white matter organization in real brains with folded cortices. Specifically, the authors present a computational model (based on the notion of tension-induced fiber growth) that predicts a predominance of tangentially oriented axons beneath cortical sulci, and then provide brain imaging evidence to show that the prediction is borne out. The idea that axons “the brain [i.e., cortex] pulls on axons” rather than vice versa, has been stated before (e.g., Holland et al. 2015), but the present authors go further by focusing on the fact that the orientation of the “pulling” forces is not always radial. We can argue over whether this insight/discovery rises to the level of a “novel paradigm” (as stated in the abstract), but it does represent a significant advance in understanding. Of course, I have a few major and several minor suggestions for improvement.

Major Issues:

- 1) I had a lot of trouble understanding Figure 2 and suspect that it can be improved.
 - (a) In panel A, I would replace “pre-folded” with “before folding” or something equivalent, since “pre-folded” means (to my mind) “already folded”.
 - (b) Panel B is where I ran into real trouble. First, I think this figure illustrates “stress-induced axon thickening” more than stress-induced elongation. I understand that in experimental studies you can get axons that become thinner when they are rapidly stretched and then return to their normal thickness as they grow. However, your text mainly refers to tension-induced axon elongation (e.g., page 7: “fibers parallel to the direction of applied tension will slowly elongate”), which to my way of thinking does not imply clearly distinct periods of stretch versus restoration/growth. As I understand your model, the tension builds gradually, rather than in sudden bursts (though I suppose buckling might cause the rates of changes in tension to vary locally). Please clarify, if possible.
 - (c) A second confusing aspect of panel B is that the middle and right rectangular slabs are shown as being equal in volume, but one is labeled as $V = V_0$ while the other is labeled $V > V_0$. After a considerable amount of time spent staring at this figure, I think you mean to illustrate that stretching the cube radially will increase the density of radially oriented axons (under a given area of cortical surface, due to what you refer to as Poisson’s effect), which will then (after the stress-induced thickening) lead to an increase in the volume fraction (but not the number) of axons that run parallel to the applied tension. I’m still not sure I understand this properly, but I think part of the problem is that I, as reader, was slow to appreciate the importance in your model of having a relatively low density of axons initially. Perhaps my somewhat inchoate reaction to your figure will help you improve it. Sorry I can’t make more specific recommendations.
 - (d) Finally, regarding panel C, it might be helpful to explain further what this graph is meant to illustrate. It might help to show on the same graph what happens to the volume fraction for the other axons (f_2); presumably it goes to zero as f_1 goes to 1 ...
- 2) I really liked how you computed local eigenvectors throughout the monkey cortex, as shown in your Figure 5, rather than merely showing an illustrative example region (as in your figure 4). I was surprised, therefore, that you resorted to a sample illustration of your data for the second part of your manuscript, where you’re describing and then testing the ellipsoid model (Fig. 6). I agree that the illustration confirms your prediction, but why didn’t you strive to develop a more objective (less prone to bias) test? I don’t want to be so presumptuous as to tell you what you should have done, but I can imagine, for example, a histogram of eigenvector angles that compares what you see when you have a ventricle versus a caudate/putamen, and/or a graph that shows how those angles vary with ventricular surface curvature. I think anything that provides a more comprehensive and objective measure of your DTI findings relative to model predictions would make the manuscript stronger. Also, I’d love to know if the “tangential” fibers along the ventricular surface are oriented parallel to the long axis of the “ellipsoid” in vivo (e.g., along the temporal

lobe) as well as in your model. I presume this is the predicted orientation, but you don't really say (other than something in the discussion about the longitudinal fasciculi).

3) I liked the findings from your exploration of parameter space and would have liked to see them spelled out a bit more and integrated better into the main paper. As it is, these findings are only presented in the supplementary figure, supplementary text (page 27, lines 16-22) and a bit of text on page 17, but I think they are more interesting and important than these placements suggest. I know there are manuscript space constraints, but perhaps a bit of reviewer-induced manuscript elongation is possible.

4) The last paragraph of the Discussion doesn't do much for me. I think that space could be better used for discussion of some of the other issues mentioned above.

Minor Issues:

page 4 line 13: You wrote: "the least restriction direction of displacement parallel to the primary fiber orientation". This is very difficult to parse and, given that this is the Introduction, a more accessible wording would be useful. At least change "restriction" to "restrictive" so that you don't have one noun modifying another.

page 9, line 4: I think a word is missing between "progression" and "subplate"

page 10: I am not clear on your distinction between subplate and inner/out fibrous layer; can you provide a reference for this distinction? I did not see one in Kostovic and Rakic (1990). This becomes important, for example, in your figure 4, where the in vivo data show a fibrous layer but your model doesn't. What are the implications of this difference between the model and the real thing? On a more practical level, I wonder how you draw the boundaries between subplate and deeper layers. One way to address this issue would be to include a supplemental figure showing variations in subplate thickness in relation to gyri (as in Kostovic and Rakic 1990) or modify Fig. 1 to indicate the extent of the subplate (maybe I'm wrong, but I think most neurobiologists would be surprised to learn that essentially all of the white matter in the illustrated sections is subplate). You might also need some additional, clarifying text somewhere.

page 10: I was a bit unclear on your statement that "In spite of the low FA value, the primary eigenvector orientation is radially oriented within the subplate". My understanding is that fractional anisotropy (FA) and the eigenvectors are related variables. Perhaps you can briefly explain how you can have a significant "primary eigenvector" at the same time as low FA (which intuitively should be high if the eigenvector is large). Also note that in Figure 4 you refer to the eigenvectors as "principal anisotropy vectors", which might confuse some readers.

It might be good somewhere to mention that an increase in fiber density within the subplate is almost certainly not JUST due to tension-induced fiber growth, but also due to the invasion of additional axons into the subplate (e.g., as this region becomes a "waiting compartment" and long descending axons come out of the cortical plate).

A question: I find it interesting that radial glia exhibit less/slower tension-induced elongation than axons. Given that these fibers are by definition radially oriented, how does their presence affect cortical folding? I suppose Baily et al. already answered that, but as the glia "fan out" in gyri, does that affect the likelihood of forming additional folds? Just food for thought.

Page 17, line 11: Change to "a novel framework"; and, yes, I like "framework" better than "paradigm" – more modest but still indicating significance!

Page 17 – lines 20-23: I haven't read the ferret tissue cutting paper in some time, but I am unclear from your text about the distinction between "tangentially subjacent" and "radially subjacent" to gyri and sulci. Perhaps an additional word or two would help the non-specialist reader understand. Also, in lines 22-23 you make it sound as if the earlier work had measured tissue compression as well as tension; are you sure? I thought they could only detect varying

degrees of tension.

Page 18, line 24: There is something missing in this sentence, or else I'm not getting the full meaning of "organization". Perhaps it is better to spell out what kind of organization has been reported.

Page 27, line 12: Insert "from" after "stem" ?

Supp. Fig. 1, caption: What is meant by "location frequency"?

Reviewer #2 (Remarks to the Author):

The manuscript by Garcia, Wang and Kroenke provides a very interesting approach to conceptualising and modelling the effect of cortical folding on white matter connectivity. The authors introduce a biomechanical model of brain development where folding of the cortical surface influences the growth and organisation of cerebral white matter fibres. Their work is also very timely. Probably the most influential hypothesis for the origin of brain folding was proposed by Van Essen in 1997. A review and extension of this hypothesis was published very recently (Van Essen, 2020). In that model axonal tension is supposed to drive cortical folding. Unlike Van Essen's model, several models proposed by different biophysicists suggest on the contrary that cortical folding is not due to cortico-cortical axonal tension, but driven by a buckling instability. The model proposed here goes one step further and suggests a mechanism through which cortical folding could direct brain connectivity, inverting the causal arrow of Van Essen's model.

Some aspects in the presentation of the model could be made more clear, however, to prevent misunderstanding and potential objections. In this work the current buckling model is complemented by equations driving the growth of white matter axons. Buckling models (for example Tallinen et al 2014) show that residual stress after folding stretches the white matter inside gyri with a direction orthogonal to the pial surface, and compresses the white matter under sulci. In Garcia's et al. model, this leads to the selective growth of fibres perpendicular to the surface in gyri, and fibres parallel to the surface in sulci.

There is however, some confusion in the presentation of the model between the aspects which are really supposed to represent observable phenomena, and those which due to simplifications required by the computational/methodological demands:

1. The model is axisymmetric, and the folding patterns look very artificial (concentric rings). This could lead some readers to dismiss this model in favour of others which produce more natural looking folding (ex., Toro 2012, Tallinen et al 2014, 2016, or even the rather unrealistic folding models based on reaction-diffusion equations which nevertheless produce more naturally looking patterns than here). It would be better to hint the reader not to focus on the shape of the folding patterns produced – whose reproduction is out of the scope of the model – and to point to the really novel point: the induction of consistent fibre orientations in correspondence with gyri and folding.

2. In the white matter the model acts on fibre orientations, however, the illustrations show streamlines (likely the integration of the direction field) which may be mistaken by fibres. Some of these fibre-looking streamlines, for example, form closed loops (blue lines in Fig. 3, for example) or do not make any contact with the grey matter. A reader could also wonder why there is no cortico-cortical fibres. These misunderstandings could lead some to unfairly dismissing the model. It should be made clear that the model aims at reproducing white matter orientation and that the streamlines are there only to support visualisation (maybe they could be completely avoided, and

show only directions as is the case for the macaque data).

I imagine that these simplifications (axisymmetric model, 'grid' streamlines) may be due to the computational demands of the model, which is reasonable, but should be clearly stated if that's the case. This should prevent readers from misinterpreting and dismissing the contribution.

Additional points follow:

4. p3 l2. Authors highlight several aspects indicating a strong link between brain folding and the organisation of white matter connectivity. The authors illustrate this point by citing results from neurological and psychiatric disorders, animal experiments and also pointing out that gyri and sulci inherently influence the length of axons. Probably U-fibres (those that link adjacent gyri by travelling parallel to sulcal fundi) should also be mentioned. U-fibres have been widely reported in several gyrencephalic species, and are not clearly explained by models such as those of Van Essen (1997, 2020) or Hilgetag & Barbas 2006 which would suggest fibres pulling together walls of the same gyrus instead. Although the streamlines illustrating the results of the model do not form U-fibres, the direction field is clearly compatible with such organisation.

5. p3 l16. The model of Richman et al. (1975, ref. 11) proposes a differential expansion of the superior versus inferior layers of the cortex, and not between the cortex versus the subplate. The models of Toro and Burnod (2005) and Tallinen et al (2016) (refs. 12, 13), such as various other similar ones propose a difference of the cortex versus all the underlying core.

6. p4 l14. The idea that cortical folding may be a causal force influencing brain organisation is novel, timely, and not widely considered in the literature. We have discussed the possibility of a causal influence of folding on cytoarchitecture and connectivity in Toro and Burnod (2005) and Toro (2012). An expanded discussion on the issue, and its importance for brain development and evolution, was further presented in Foubet et al (2019) and Heuer and Toro (2019). The authors may find in those references some ideas complementary to theirs.

7. Figure 1c may be misleading. Welker (1990), Ronan and Fletcher (2014), De Juan Romero et al (2015) among others have proposed a model in which different cortical regions grow at different rates. For example, a genetic pre-patterning of the cortex would determine which regions grow more than others to become gyri. This model is sometimes referred as the "differential expansion" hypothesis: what is differentially expanding is not the cortex over the substrate (constrained growth), but one cortical region relative to another. Figure 1c illustrates growth in the cortex with longer arrows in gyri compared with sulci, which may lead some readers to think they are actually referring to "differential expansion" and not to "constrained growth". Maybe arrows of the same length could be used throughout?

8. p7 l9. What was Poisson's ratio for the fibres (in steady state)? One could imagine that shrinking due to Poisson's effect could trigger growth in fibre caliber to compensate stress, leading to a Poisson's ratio of ~ 0 instead of ~ 0.5 used for soft elastic tissue?

9. p7 l13. In the model, under the action on stress, an element of volume will end up having fibres of a single orientation. However, in reality multiple directions often coexist (which is a well recognised problem for diffusion weighted imaging tractography). Van Wedeen et al. (2012) even proposed a conceptual model where white matter axons would always belong to 1 of 3 locally orthogonal orientations, based on DWI tractography data supported by histological data. How are these observations compatible with the model?

10. p8 l7. In the model folding was triggered by making a small region grow 5% faster than the rest of the cortex (which reminds to some extent the differential expansion hypotheses of Welker, Ronan, Borrell,...). Was this necessary to trigger folding? Wouldn't the addition of a very small, homogeneous, geometric noise be sufficient? As far as I know, in most buckling models numerical noise is sufficient to trigger folding. Was the idea to control the pattern of folding more precisely? What is the size of this small region? What is its shape? Was the rate of growth of this region 5% larger only at the beginning of the simulation or throughout the simulation?

11. p20 l3. Why cortical growth in the model is only tangential? Cortical thickness does increase through the period of cortical folding (likely due to the neuropil development, dendrites, axonal collaterals,...). In ferrets, for example, cortical thickness goes from <1mm at before folding (P4) to ~2mm at ~2 weeks (adult-like folding pattern). If growth is isotropic instead of just tangential such x2 increase in length, would translate to a x8 increase in volume, which is about the increase of brain volume observed within that period? Is the assumption that radial cortical growth is only the passive response to the pull of the white matter core and that all active growth is exclusively tangential?

12. Because of the remodelling of the white matter fibres, I can't get a general idea of its overall elasticity, to be able to compare this model with previous models. In the context of the literature on buckling of bi-layers, does the present model correspond more to a wrinkling regime such as the models of Budday et al (2014), Bayly et al (2013) where the top layer is stiffer than the bottom one, or a creasing regime as strongly advocated by Tallinen, Mahadevan, et al (2014, 2016), where the elasticity in both tissues is similar?

13. p17 l10: "patterns *of* cortical"

14. It would be very helpful to made the source code for the simulation available (as is for example the case with the models of Toro et al. or Tallinen et al.). This could facilitate replication as well as future comparisons and extensions of the model.

R. Toro

Reviewer #3 (Remarks to the Author):

In this paper, the authors propose a new model that takes the volume fraction of axonal fiber into account. The model's outcomes – the predicted fiber organizations in the underlying white matter – are compared qualitatively against the high-resolution ex vivo diffusion tensor imaging of the developing rhesus macaque brain. The authors conclude that the proposed model supports a causative role for folding in the stereotyped fiber organization beneath gyri and sulci. While the model builds upon the authors' previous work (Bayly et al., 2013), this manuscript's novelty lies in incorporating the volume fraction of the evolving axonal fiber in the anisotropic material model.

While the paper is extremely interesting, I believe that there are several unanswered questions regarding the proposed constitutive model. Before this manuscript is ready for publication, the authors need to address the following concerns:

1. Compared to the evolution equation in their previous work (Bayly et al., 2013), the evolution equation proposed here contains the volume fraction f_i in front. This modification is briefly explained, but not fully justified. It is not immediately apparent to me that the growth rate (as opposed to the total volume change) should depend on the volume fraction of fibers. In addition to a more thorough justification of this governing equation, it would be helpful to further illustrate its effect under simple loading conditions. For instance, Fig. 2 could also contain graphs showing the total volume and the imposed stretch on the hypothetical tissue volume element with fibers.

2. While the volume fraction of axonal fiber changes in response to the stress, it does not in turn affect the stiffness of the material. Mechanical tests have shown that white matter has pronounced anisotropic and stretch-locking behaviors (Garcia-Gonzalez et al., 2018). The authors should consider this in their model, including a non-linear term in the mechanical free energy (Eq. 4) to capture this behavior.

3. This study would be more compelling if the effect of various initial fiber organizations were investigated and illustrated. For instance, using a different coordinate system (such as Cartesian instead of spherical) for the initial fiber directions, or assigning slightly different initial fiber densities to certain direction(s) (instead of the predominantly radial orientation shown here). Are the findings robust to changes in initial alignment?

4. In Figure 4d, fibers appear to be highly aligned under both the gyral crown and sulcal fundi, but there is an observable difference in FA value between the two regions. Similarly, in the macaque brains, the FA value does not change significantly in time or vary significantly between gyri and sulci. This difference should be discussed further.

5. The choice of exponential cortical growth, while based on data from macaque development, is unusual. Most, if not all, of the studies used for comparison in this manuscript used linear growth. Considering that a major point is comparisons between axonal growth rates in these studies, this presents a significant problem, as axonal and cortical growth rates can really only be understood relative to each other. This choice should be justified further, and its affect on the relationship between axonal and cortical growth rates should be discussed and, if possible, quantified.

Additionally, I believe the paper would be strengthened by addressing the following comments:

1. The term 'viscoelastic growth' is used in several places to refer to the behavior of the subplate, and it is not clear what is meant by this. Growth represents the addition of new mass while viscoelasticity is the result of microstructural changes. While the resulting behavior appears similar in some contexts, the two processes are not the same, and this distinction should be made more clearly to avoid confusing readers.

2. The authors explain their calculations of the stress-dependent fiber elongation parameter at length. However, consider that the experiments analyzed in Holland et al. 2015 include stretches of almost 800% in an hour, which is hardly physiological. Assuming that most experiments rely on similarly aggressive stresses and stretches, this might suggest that parameters based on in vitro experiments represent upper bounds of what axons are capable of, rather than the quantification of physiological behavior.

3. The evolution equation (Eq. 2) also allows for tissue shrinkage when $\sigma_i < \sigma_0$, or when the tissue is in compression. This is alluded to in lines 11 and 12 but should be made explicit.

4. The numerator and denominator in Eq. 3 appear to be (unitless) volume ratios, not volumes. Perhaps the second term would be better written as $J_{\text{fibers},i} / J_{\text{total}}$.

References

PV Bayly, RJ Okamoto, G Xu, Y Shi, and LA Taber. A cortical folding model incorporating stress-dependent growth explains gyral wavelengths and stress patterns in the developing brain. *Physical biology*, 10(1):016005, 2013.

Daniel Garcia-Gonzalez, A Jerusalem, Sara Garzon-Hernandez, R Zaera, and A Arias. A continuum mechanics constitutive framework for transverse isotropic soft tissues. *Journal of the Mechanics and Physics of Solids*, 112:209–224, 2018.

Reviewer #4 (Remarks to the Author):

The authors propose that stress induced by cortical growth and folding might explain the white matter configurations seen in the underlying white matter. In particular, they argue that stress-induced fibre growth might explain the tangential orientations in the white matter underlying sulci and the radial orientations in the white matter underlying gyral crowns. These patterns are shown in ex-vivo macaque DTI data for comparison. This is an interesting proposal, however I am confused about the underlying mechanics.

Model queries:

1. Could the authors clarify what is actually happening during the initial tissue stretch and the subsequent fiber elongation phase shown in Figure 2. How can the tissue stretch without the axons within it elongating (or breaking)? And if the fibres elongate during the tissue deformation phase,

what is actually the growth in the second phase? The illustration in Figure 2 seems to suggest that during the "fiber elongation" phase it is actually the axonal diameters that change, which does not appear in line with my (admittedly very cursory reading) of the papers describing fiber elongation under stress. It would be very helpful to spell out explicitly what the model implies for what happens with the axonal length, axonal diameter, and number of axons under stress, keeping in mind that the tissue cannot deform without axonal elongation/shrinkage.

2. The model does not appear to take into account extra-cellular space. While stretching tissue in one dimension would reduce the density of cells and axons not aligned with the stretch, the extra-axonal space will increase as the tissue volume increases. This would prevent the fibre volume fraction from going to one (Figure 2). This assumption of no extra-cellular space greatly influences the total fibre density increase seen in Figure 3, which would instead remain roughly constant if the extra-axonal space was extra-cellular rather than cellular.

3. The result of the simulations seem to imply that the increase in the white matter's fractional anisotropy during development can be explained by tissue deformations, without the traditional explanations of new axon formation and myelination. Do the authors actually think this is the case? If so, it would be good to actually discuss this surprising result in the discussion.

Minor comments:

1. When introducing the mechanical stress σ it would be very helpful to mention that this is related to the elastic deformation tensor F^* through equations given in the methods section, so the reader knows where to find the remaining equations.

2. It would be good to mention that the layer selected in Figure 6 is supposed to represent the "superficial subplate" in the caption or the main text rather than only in the methods section.

3. The fibre orientations in Figures 4-6 are hard to see. I would suggest to make the figures bigger and/or color code the fibre orientations based on how radial orientations. The latter would make it a lot easier to see the main point of these figures at a glance.

4. Page 13; line 6: insert "of" in "features subcortical fiber organization"

5. Page 13; line 27: "FA increased" should be "increased FA"

6. Please define J^* in equation 4

RESPONSE TO REVIEWER COMMENTS

We would like to thank the Reviewers for their supportive and insightful comments regarding this paper. We have made substantial changes to the manuscript to address the issues raised, and we believe the revised version is much improved.

Some of the suggested additions are items we had originally hoped to include but were unable to accommodate within journal space constraints. We appreciate the challenge and encouragement from Reviewers to include these items, as we feel they provide additional insights and further illustrate the importance of our proposed model. However, to accommodate these additions in the main text where possible while remaining within the Nature Communications text limits, the Theory described in our original manuscript has been moved to Methods, and some of the original Discussion has been streamlined. Additional requests are addressed in the expanded Supplement where appropriate.

Reviewer #1 (Remarks to the Author):

I think this is a well written manuscript describing an interesting set of models and ex vivo observations that, together, make a substantial contribution to the literature on cortical folding. The authors have presented evidence for a novel idea, namely that mechanical forces created by cortical folding (buckling forces) affect white matter organization in real brains with folded cortices. Specifically, the authors present a computational model (based on the notion of tension-induced fiber growth) that predicts a predominance of tangentially oriented axons beneath cortical sulci, and then provide brain imaging evidence to show that the prediction is borne out. The idea that axons “the brain [i.e., cortex] pulls on axons” rather than vice versa, has been stated before (e.g., Holland et al. 2015), but the present authors go further by focusing on the fact that the orientation of the “pulling” forces is not always radial. We can argue over whether this insight/discovery rises to the level of a “novel paradigm” (as stated in the abstract), but it does represent a significant advance in understanding. Of course, I have a few major and several minor suggestions for improvement.

We thank the reviewer for their positive comments. In addition to addressing issues below, we have adjusted our terminology to avoid the term “novel paradigm”. (We have also minimized use of the terms novel, new, first, etc. throughout the manuscript, in accordance with additional guidance on Nature formats.)

Major Issues:

1) I had a lot of trouble understanding Figure 2 and suspect that it can be improved.

In our revised manuscript, Figure 2 has been modified and improved significantly. We have also re-arranged this content by (1) moving an improved illustration of the original panel A to Figure 1 and (2) moving the rest of Figure 2 to Methods (now Fig. 8). We feel both of these new locations are more appropriate for the points made in these images. Moving Figure 2 and accompanying text to the Methods section offers two benefits:

1. This avoids confusing readers up front with details of the implementation, as we believe the basic points can be appreciated in the updated Figs. 1-2.

2. This provides more space in the main text to adequately describe this process and address other reviewer comments. (Methods do not fall under the space limitations for the main text.)

(a) In panel A, I would replace “pre-folded” with “before folding” or something equivalent, since “pre-folded” means (to my mind) “already folded”.

For the new panel in Figure 1, “pre-folded” terminology is not used.

(b) Panel B is where I ran into real trouble. First, I think this figure illustrates “stress-induced axon thickening” more than stress-induced elongation. I understand that in experimental studies you can get axons that become thinner when they are rapidly stretched and then return to their normal thickness as they grow. However, your text mainly refers to tension-induced axon elongation (e.g., page 7: “fibers parallel to the direction of applied tension will slowly elongate”), which to my way of thinking does not imply clearly distinct periods of stretch versus restoration/growth. As I understand your model, the tension builds gradually, rather than in sudden bursts (though I suppose buckling might cause the rates of changes in tension to vary locally). Please clarify, if possible.

The Reviewer is correct that this process occurs continuously, with very low forces and slow elongation in response, rather than in a dramatic burst as illustrated. This is more clearly illustrated in the revised Figure (now Fig 8) and described in text:

The consequences of these governing equations are illustrated in Fig. 8, considering a simple block of tissue subjected to sustained tension in one direction and zero stress in perpendicular directions. Based on histological studies suggesting that axonal fibers of the early subplate are sparse⁸, fiber volume fractions in this example are defined to account for only a small fraction of the overall tissue volume at the outset ($f_1 = f_2 = f_3 = 0.1$ at $t=0$). As fiber components parallel to the direction of tension elongate, growth is observed in the direction of tension (Fig. 8a). Since fibers are only defined to elongate, not to increase in cross-section (caliber), no tissue growth is observed in directions perpendicular to the applied tension.

(c) A second confusing aspect of panel B is that the middle and right rectangular slabs are shown as being equal in volume, but one is labeled as $V=V_0$ while the other is labeled $V>V_0$. After a considerable amount of time spent staring at this figure, I think you mean to illustrate that stretching the cube radially will increase the density of radially oriented axons (under a given area of cortical surface, due to what you refer to as Poisson’s effect), which will then (after the stress-induced thickening) lead to an increase in the volume fraction (but not the number) of axons that run parallel to the applied tension. I’m still not sure I understand this properly, but I think part of the problem is that I, as reader, was slow to appreciate the importance in your model of having a relatively low density of axons initially. Perhaps my somewhat inchoate reaction to your figure will help you improve it. Sorry I can’t make more specific recommendations.

Again, the Reviewer is correct in the intended interpretation, and we appreciate the suggestion to make this clearer for all readers. In the revised Figure (now Fig 8), we have removed the “V” labels and explained changes in volume via subsequent panel illustrating the volume associated with each component over time. Based on the comments of multiple reviewers, we have also simplified our illustrated case. We no longer discuss a true “uniaxial” stretch that induces shrinkage in non-stretched direction (Poisson’s effect), but rather a theoretical 1D stretch case in which stresses in the other directions are assumed to be zero.

As fibers increase in length while maintaining their caliber, the corresponding fiber volume increases (Fig. 8b), and the relative volume fraction of fibers in this direction slowly approaches 1 as time approaches infinity (Fig. 8c). For any direction of sustained tension, the above equations also suggest an increase in total fiber volume fraction approaching 1 as time approaches infinity (Fig 8c).

(d) Finally, regarding panel C, it might be helpful to explain further what this graph is meant to illustrate. It might help to show on the same graph what happens to the volume fraction for the other axons (f_2); presumably it goes to zero as f_1 goes to 1 ...

We appreciate the reviewer's suggestion regarding panel C. We have added additional lines to the new Figure that describe what happens for each tissue component, in terms of both volume and volume fraction. We believe this excellent suggestion adds substantial clarity to the figure and theory described in this paper.

2) I really liked how you computed local eigenvectors throughout the monkey cortex, as shown in your Figure 5, rather than merely showing an illustrative example region (as in your figure 4). I was surprised, therefore, that you resorted to a sample illustration of your data for the second part of your manuscript, where you're describing and then testing the ellipsoid model (Fig. 6). I agree that the illustration confirms your prediction, but why didn't you strive to develop a more objective (less prone to bias) test? I don't want to be so presumptuous as to tell you what you should have done, but I can imagine, for example, a histogram of eigenvector angles that compares what you see when you have a ventricle versus a caudate/putamen, and/or a graph that shows how those angles vary with ventricular surface curvature. I think anything that provides a more comprehensive and objective measure of your DTI findings relative to model predictions would make the manuscript stronger.

Also, I'd love to know if the "tangential" fibers along the ventricular surface are oriented parallel to the long axis of the "ellipsoid" in vivo (e.g., along the temporal lobe) as well as in your model. I presume this is the predicted orientation, but you don't really say (other than something in the discussion about the longitudinal fasciuli).

We appreciate the reviewer's comments on Fig. 5 and suggestion to include a similar analysis of Fig. 6. In the revised manuscript, we have included an equivalent theta map corresponding to the ventricular surface that supports our model. A similar map was unfortunately not feasible for the tissue-filled region, due to the presence of specialized structures including the internal capsule that complicate the picture. (While trends generally support our model, this level of detail raises many more questions relating to these specific structures.) We also confirm that the "tangential" fibers are parallel to the long axis of the ventricle ellipsoid. This is discussed briefly in Results:

To confirm these predictions as they relate to brain geometry, model fiber orientations were again compared to high resolution DTI in the rhesus macaque. As shown in Fig. 7, increased FA and tangentially-oriented primary eigenvectors (running parallel to the "long axis" of the ventricle) were visible near the ventricle in the G85 macaque brain, consistent with previous reports³². Furthermore, tangential organization along the ventricle was most apparent in the low curvature portion (along its length), suggesting that organization near the poles may occur later, consistent with the predicted time course in Fig. 6b.

By contrast, models with a solid inner core generally maintained radial organization in deeper layers, with tangential organization only appearing in a mid-depth location for ellipsoidal models. This radial organization could relate to projection fibers that extend from deeper structures into the cortex, and similar organization was observed in select brain regions (Fig. 7a, red inset). However, this prediction

could not be robustly confirmed with our current dataset, as diffusion anisotropy in white matter near striatal nuclei is dominated by early-developing fiber tracts such as the internal capsule and corpus callosum. Presumably, additional biological mechanisms such as molecular axon guidance cues influence the organization of these white matter structures independently of mechanical forces.

To avoid an overly busy figure and adequately address other reviewer comments, we have split the original Fig. 6 into 2 separate figures: Fig 6 focused on the model effects of ellipsoid elongation and ventricle vs no ventricle, and Fig. 7 focused on more detailed and quantitative comparison of the G85 periventricular orientations. The revised Fig. 7 includes a new reconstruction of the G85 ventricle, with theta map representing the periventricular orientations plotted on the surface, paralleling our display of subplate orientations in Fig. 5. The trends in orientation are also displayed as a function of distance along the long axis of the ellipsoid and as a function of minimum curvature.

As suggested by the reviewer, tangential fibers are oriented parallel to the long axis of the ellipsoid. This can be observed in the original tick mark orientation plots, as a primary orientation “into the page” would be observed as a dot (or very short line). We have attempted to clarify and further highlight this point in our revised manuscript, aided by discussion of these new quantification approaches.

3) I liked the findings from your exploration of parameter space and would have liked to see them spelled out a bit more and integrated better into the main paper. As it is, these findings are only presented in the supplementary figure, supplementary text (page 27, lines 16-22) and a bit of text on page 17, but I think they are more interesting and important than these placements suggest. I know there are manuscript space constraints, but perhaps a bit of reviewer-induced manuscript elongation is possible.

Thank you for your comments regarding this point. We also feel these points are very interesting and worth inclusion in the main paper, but they were initially relegated to Supplement based on space constraints. Moving Fig. 2 and accompanying text to Methods has been instrumental in managing word count limits. In our revised manuscript, we have refined our supplementary Figure and discussion on this topic, now included as a new Results section in the main text (Results heading 2: **Initial fiber properties influence cortical fold morphology**), immediately following the initial/introductory results section of related to our model (Results heading 1: **Folding stresses induce patterns in subplate fiber networks**)

4) The last paragraph of the Discussion doesn't do much for me. I think that space could be better used for discussion of some of the other issues mentioned above.

Thank you for this helpful feedback - we have removed this paragraph from the revised manuscript.

Minor Issues:

page 4 line 13: You wrote: “the least restriction direction of displacement parallel to the primary fiber orientation”. This is very difficult to parse and, given that this is the Introduction, a more accessible wording would be useful. At least change “restriction” to “restrictive” so that you don't have one noun modifying another.

We agree this was clumsily worded. We have changed the sentence to simply state

Water diffusion in the subplate proximal to the cortical plate was indeed found to exhibit anisotropy, including spatial and temporal trends in agreement with current simulations.

page 9, line 4: I think a word is missing between “progression” and “subplate”

Thank you for catching this. The missing word has been added.

page 10: I am not clear on your distinction between subplate and inner/out fibrous layer; can you provide a reference for this distinction? I did not see one in Kostovic and Rakic (1990). This becomes important, for example, in your figure 4, where the in vivo data show a fibrous layer but your model doesn't. What are the implications of this difference between the model and the real thing? On a more practical level, I wonder how you draw the boundaries between subplate and deeper layers. One way to address this issue would be to include a supplemental figure showing variations in subplate thickness in relation to gyri (as in Kostovic and Rakic 1990) or modify Fig. 1 to indicate the extent of the subplate (maybe I'm wrong, but I think most neurobiologists would be surprised to learn that essentially all of the white matter in the illustrated sections is subplate). You might also need some additional, clarifying text somewhere.

We thank the reviewer for pointing out these issues. To address the inquiry, we have detailed how different tissue zones are assigned in our prior work (Wang et al., 2015, and Table 1 of that reference directly compares 5 resolvable tissue zones in MR data to lamina defined by other authors). That reference also explicitly recognizes limitations of MRI that arise from low image resolution compared to microscopy.

After taking this input, we believe it adds more confusion than clarity to attempt to delineate different non-cortical lamina in Figure 4, where the primary consideration is cortical plate vs. sub-cortical tissue. We have therefore colored eigenvectors for all tissue outside of the cortical plate yellow, and removed reference in the text to specific lamina. We have also added a citation to Smart et al., (2002) along with Kostovic and Rakic, (1990) in the second paragraph of the Introduction, for a reference that includes inner/outer fibrous layers.

page 10: I was a bit unclear on your statement that “In spite of the low FA value, the primary eigenvector orientation is radially oriented within the subplate”. My understanding is that fractional anisotropy (FA) and the eigenvectors are related variables. Perhaps you can briefly explain how you can have a significant “primary eigenvector” at the same time as low FA (which intuitively should be high if the eigenvector is large). Also note that in Figure 4 you refer to the eigenvectors as “principal anisotropy vectors”, which might confuse some readers.

To address the reviewer's inquiry, all 3 diffusion tensor eigenvectors have the same (unit) length. Fractional anisotropy reflects differences in the sizes of the eigenvalues (not eigenvectors) for a given diffusion tensor. We have removed the clause “In spite of the low FA value” from the text to eliminate this source of confusion. We also agree with the reviewer that the terminology “principal anisotropy vector” is confusing and have changed that to the more customary “diffusion tensor principal eigenvector”.

It might be good somewhere to mention that an increase in fiber density within the subplate is almost certainly not JUST due to tension-induced fiber growth, but also due to the invasion of additional axons into the subplate (e.g., as this region becomes a “waiting compartment” and long descending axons come out of the cortical plate).

We agree this is an important point to incorporate that was not sufficiently emphasized in the prior manuscript draft. We have added the following paragraph to the Discussion:

It is important to acknowledge that the current model only approximates growth by focusing on specific developmental mechanisms, such as tension-induced fiber elongation and cortical expansion. For example, it does not account for axon growth resulting from other biological mechanisms such as well-established molecular axon guidance factors⁵⁴. Indeed, prior to cortical folding, afferent axons from cortical and subcortical structures form synapses within the subplate, and efferent axons project from the cortex based on previously described non-mechanical cues. The present model's failure to reproduce the complex organization observed near solid deep brain structures, such as the striatum, highlights the need for additional organizing factors to fully understand brain development. Furthermore, the role of mechanical feedback may extend beyond axon elongation. The influence of feedback on biological processes such as neurite branching and cell proliferation^{48,45} has provided increasing support for the idea that mechanical factors act in concert with other factors (eg., innate genetic determinants) during cortical morphogenesis^{44,55}. These factors will be important to consider in future models of cortical and subcortical development.

A question: I find it interesting that radial glia exhibit less/slower tension-induced elongation than axons. Given that these fibers are by definition radially oriented, how does their presence affect cortical folding? I suppose Bayly et al. already answered that, but as the glia "fan out" in gyri, does that affect the likelihood of forming additional folds? Just food for thought.

We agree that this is a very interesting point worth pursuing in future studies. Our current model would suggest a role in ultimate fold morphology, as well as a potential role in symmetry breaking/fold patterning if different distributions are observed in different locations. However, very little data is available for the elongation of radial glial processes in response to mechanical stimuli. Another complication is that radial glia become depleted as they differentiate into other cell types while the cortex folds. In the absence of quantitative data characterizing these processes, we hesitate to make specific claims about their influence. However, we believe our streamlined Discussion helps to address this point:

By applying axon elongation rates to only fiber components, a relatively small proportion of early subcortical tissue, net growth of the subplate was reduced to a level that facilitates folding. Still, we note that only rates at the lower end of the reported range were capable of producing a folded cortex. This result may suggest a slower response of neurites under *in vivo* conditions during the period of folding, with *in vitro* experimental conditions (including relatively rapid stretches compared to those anticipated during brain development) representing the upper bound of axon elongation behavior. Moreover, simulations presented here do not distinguish fiber contributions from neuronal and non-neuronal sources (neurites versus radial glial scaffolds). Radial glial processes have been observed to extend at rates comparable to axonal growth cones⁵⁰, and astrocyte processes designed to mimic radial glial processes have recently been shown to elongate under sustained tension *in vitro*, although at a slower rate³⁹. The need for values on the lower end of the experimentally-reported range may therefore reflect an early predominance of slower-growing fibers such as radial glial scaffolds³⁹. Additional studies are needed to clarify the rates of tension-induced elongation of various fiber types in the context of brain folding, as well as the precise volume fractions of these components during development, in relevant gyrencephalic model systems such as the developing ferret or rhesus macaque.

Page 17, line 11: Change to "a novel framework"; and, yes, I like "framework" better than "paradigm" – more modest but still indicating significance!

All instances of "paradigm" have been removed from the revised manuscript, replaced with "framework" or "hypothesis".

Page 17 – lines 20-23: I haven't read the ferret tissue cutting paper in some time, but I am unclear from your text about the distinction between “tangentially subjacent” and “radially subjacent” to gyri and sulci. Perhaps an additional word or two would help the non-specialist reader understand . Also, in lines 22-23 you make it sound as if the earlier work had measured tissue compression as well as tension; are you sure? I thought they could only detect varying degrees of tension.

We have adjusted the text to avoid confusion. The intention of these statements was to describe how fibers are oriented tangentially beneath sulci and radially beneath gyri. To clarify our meaning in this respect, we have replaced the word “subjacent” with “below” or “beneath”

You are correct that earlier work focused on measuring tissue tension, as observed by formation of an opening during tissue cutting experiments. Though the spaces that do not exhibit tension must, by the same logic, indicate either zero stress or compression. In lines 22-23, we are not discussing these experiments, but rather the theoretical stress state implied by “pushing” forces. We have also attempted to clarify the text to avoid this confusion.

First, the pattern of subplate tension predicted by cortical expansion-driven models (greatest along tangential directions **beneath** sulci and radial directions **beneath** gyri) is consistent with the directions of **tension previously observed in the developing ferret brain**¹⁴.

Page 18, line 24: There is something missing in this sentence, or else I'm not getting the full meaning of “organization”. Perhaps it is better to spell out what kind of organization has been reported.

We have adjusted this statement for clarity, changing “tangential organization” to “predominance of tangentially-oriented tissue components.”

Page 27, line 12: Insert “from” after “stem” ?

We thank the reviewer for catching this typo. This has been corrected.

Supp. Fig. 1, caption: What is meant by “location frequency”?

We thank the reviewer for catching this typo. This has been corrected in what is now Fig. 3 of the main text.

Reviewer #2 (Remarks to the Author):

The manuscript by Garcia, Wang and Kroenke provides a very interesting approach to conceptualising and modelling the effect of cortical folding on white matter connectivity. The authors introduce a biomechanical model of brain development where folding of the cortical surface influences the growth and organisation of cerebral white matter fibres. Their work is also very timely. Probably the most influential hypothesis for the origin of brain folding was proposed by Van Essen in 1997. A review and extension of this hypothesis was published very recently (Van Essen, 2020). In that model axonal tension is supposed to drive cortical folding. Unlike Van Essen's model, several models proposed by different biophysicists suggest on the contrary that cortical folding is not due to cortico-cortical axonal tension, but driven by a buckling instability. The model proposed here goes one step further and suggests a mechanism through

which cortical folding could direct brain connectivity, inverting the causal arrow of Van Essen's model.

Some aspects in the presentation of the model could be made more clear, however, to prevent misunderstanding and potential objections. In this work the current buckling model is complemented by equations driving the growth of white matter axons. Buckling models (for example Tallinen et al 2014) show that residual stress after folding stretches the white matter inside gyri with a direction orthogonal to the pial surface, and compresses the white matter under sulci. In Garcia's et al. model, this leads to the selective growth of fibres perpendicular to the surface in gyri, and fibres parallel to the surface in sulci.

There is however, some confusion in the presentation of the model between the aspects which are really supposed to represent observable phenomena, and those which due to simplifications required by the computational/methodological demands:

1. The model is axisymmetric, and the folding patterns look very artificial (concentric rings). This could lead some readers to dismiss this model in favour of others which produce more natural looking folding (ex., Toro 2012, Tallinen et al 2014, 2016, or even the rather unrealistic folding models based on reaction-diffusion equations which nevertheless produce more naturally looking patterns than here). It would be better to hint the reader not to focus on the shape of the folding patterns produced – whose reproduction is out of the scope of the model – and to point to the really novel point: the induction of consistent fibre orientations in correspondence with gyri and folding.

We thank the reviewer for raising this point. We have added several statements throughout the text to emphasize the novel points and de-emphasize the concentric ring folding patterns produced by our axisymmetric model:

- To restrict fold formation to the plane of interest, a two-dimensional axisymmetric model was selected. While this model cannot replicate the branched folding observed in full three-dimensional models¹⁶, restriction of folding to concentric rings facilitates precise quantification of stresses in the cross-sectional plane of each fold as well as along the length of each fold.
- Furthermore, folding-induced patterns of stress led to changes in the distribution of fibers, resulting in increased tangential fiber volume fraction beneath sulci and increased radial fiber fraction beneath sulci (Fig. 2d). This stereotyped pattern beneath folds is generally consistent with white matter organization observed in the adult human brain (Fig. 1d)^{11,12}.

Prompted your later comment, as well as the requests by other reviewers, we also highlight aspects of our model that are more realistic than the “creasing” models by Tallinen et al. (2014, 2016) and other elastic/hyperelastic models that focus solely on achieving complex geometries at later stages of folding (p10, paragraph 2) .

2. In the white matter the model acts on fibre orientations, however, the illustrations show streamlines (likely the integration of the direction field) which may be mistaken by fibres. Some of these fibre-looking streamlines, for example, form closed loops (blue lines in Fig. 3, for example) or do not make any contact with the grey matter. A reader could also wonder why there is no cortico-cortical fibres. These misunderstandings could lead some to unfairly dismissing the model. It should be made clear that the model aims at reproducing white matter orientation and that the streamlines are there only to support visualisation (maybe they could be completely avoided, and show only directions as is the case for the macaque data). I imagine that these simplifications (axisymmetric model, ‘grid’ streamlines) may be due to the

computational demands of the model, which is reasonable, but should be clearly stated if that's the case. This should prevent readers from misinterpreting and dismissing the contribution.

This comment is also much appreciated. In the revised manuscript, we have adjusted the visualization in panel D to show local "+"s indicating fiber density in each direction, rather than connected streamlines. This is further described and clarified in the caption:

In (d), crosses (+) are used to visualize fiber volume fraction in terms of radial and tangential directions. Fiber volume fraction is represented by color (radial fibers in red, tangential fibers in blue), such that a dark red region indicates increased radial fibers per unit volume, dark blue indicates increased tangential fibers per unit volume, and light color indicates relatively low fiber volume fraction.

Additional points follow:

4. p3 I2. Authors highlight several aspects indicating a strong link between brain folding and the organisation of white matter connectivity. The authors illustrate this point by citing results from neurological and psychiatric disorders, animal experiments and also pointing out that gyri and sulci inherently influence the length of axons. Probably U-fibres (those that link adjacent gyri by travelling parallel to sulcal fundi) should also be mentioned. U-fibres have been widely reported in several gyrencephalic species, and are not clearly explained by models such as those of Van Essen (1997, 2020) or Hilgetag & Barbas 2006 which would suggest fibres pulling together walls of the same gyrus instead. Although the streamlines illustrating the results of the model do not form U-fibres, the direction field is clearly compatible with such organisation.

We thank the reviewer for this valuable suggestion. We agree with these points related to U-fibers, and observation of these were the original inspiration for this project. We have modified our manuscript to more clearly reference U fibers throughout the text. An illustration is provided in Figure 1D. In addition, our revised manuscript more explicitly describes and discusses U-fibers.

- **Intro:** In the adult human brain, short association fibers that connect adjacent gyri (U-fibers) exhibit highly stereotyped organization with respect to fold morphology (Fig. 1d), and long association fibers (such as longitudinal fasciculi) are highly consistent across individuals^{11,12}.
- **Fig 1d:** Image of human brain from Williams et al²⁸ illustrates predominant organization observed in the adult white matter, including radial organization beneath gyri and tangential organization beneath sulci. Short association fibers bridging adjacent gyri (U-fibers) are indicated with asterisks (*).
- **Discussion:** Observations are also consistent with tractography studies using fetal human DTI⁹ that have reported fibers sweeping tangentially beneath sulci and radially into gyral crowns, similar to U-fibers observed in adult white matter (Fig. 1d). By contrast, the hypothesis that axons pull the cortex to form gyri, which would predict a reversed pattern of predominant axon orientations^{6,13,48}.

5. p3 I16. The model of Richman et al. (1975, ref. 11) proposes a differential expansion of the superior versus inferior layers of the cortex, and not between the cortex versus the subplate. The models of Toro and Burnod (2005) and Tallinen et al (2016) (refs. 12, 13), such as various other similar ones propose a difference of the cortex versus all the underlying core.

We thank the reviewer for this correction. For simplicity, we lumped these together in the introduction, but we recognize that this is an important distinction. To more accurately group these distinct models, we have replaced "underlying subplate" with "underlying tissue" where necessary. In other places, we have removed ref 11 from the grouping as appropriate.

6. p4 l14. The idea that cortical folding may be a causal force influencing brain organisation is novel, timely, and not widely considered in the literature. We have discussed the possibility of a causal influence of folding on cytoarchitecture and connectivity in Toro and Burnod (2005) and Toro (2012). An expanded discussion on the issue, and its importance for brain development and evolution, was further presented in Foubet et al (2019) and Heuer and Toro (2019). The authors may find in those references some ideas complementary to theirs.

This comment has led us to appreciate the theme developed in the specified publications. Specifically that mechanical factors represent a class of developmental mechanisms that, together with genetic and activity-dependent factors, are responsible for folding of the cerebral cortex. We have cited this literature within the modified Discussion:

- Paragraph 1: Recent experimental and computational advances have increased recognition for the role of mechanics, alongside genetic and neural activity-dependent factors, in development of the cerebral cortex^{17,44,45}.
- Paragraph 5: Finally, simulations suggest that a reduction in initial fiber density will also produce increased gyrification (Fig. 3b, right). This is consistent with experimental studies in which early disruption of subcortical connectivity (reduction of intact fibers) led to an overproduction of cortical folds^{5,52,53}. Alternatively, as suggested by previous work¹⁷, disruption of connectivity may alter a complex array of developmental mechanisms, such as cortical growth, which could also alter the observed pattern of folding.
- Paragraph 6: The influence of feedback on biological processes such as neurite branching and cell proliferation^{48,45} has provided increasing support for the idea that mechanical factors act in concert with other factors (eg., innate genetic determinants) during cortical morphogenesis^{44,55}.

7. Figure 1c may be misleading. Welker (1990), Ronan and Fletcher (2014), De Juan Romero et al (2015) among others have proposed a model in which different cortical regions grow at different rates. For example, a genetic pre-patterning of the cortex would determine which regions grow more than others to become gyri. This model is sometimes referred as the “differential expansion” hypothesis: what is differentially expanding is not the cortex over the substrate (constrained growth), but one cortical region relative to another. Figure 1c illustrates growth in the cortex with longer arrows in gyri compared with sulci, which may lead some readers to think they are actually referring to “differential expansion” and not to “constrained growth”. Maybe arrows of the same length could be used throughout?

We agree with the reviewer’s comments related to Figure 1, and to minimize confusion at the outset of our manuscript, we have adjusted the length of the arrows to be the same in Fig 1c.

However, as discussed in comment 10 below, our main model does consider a possible relationship in which “differential expansion” acts in concert with constrained growth. As alluded in Toro and Burnod (2005), now cited, differential expansion can be viewed as the initial perturbation to bias the location of folds:

Intro:

In light of these physical measures, recent studies have focused on a third potential mechanism of folding in which cortical expansion, constrained by slower growth of the underlying subcortical tissue, leads to mechanical buckling (wrinkling or creasing)¹⁵⁻¹⁷. Constrained cortical growth is commonly considered to be uniform across the cortex, though differential rates have also been proposed to lead to preferential formation of gyri in specific areas¹⁸⁻²⁰. Critically, computational simulations of either case predict tissue stresses consistent with experimental observations^{14,21}.

Results:

To break symmetry and induce the first fold, a small region of the cortex was defined to grow 5% faster than adjacent cortex. This perturbation resembles theories of differential cortical expansion, in which local or regional variations in cortical growth have been proposed to control the formation of primary gyri¹⁸⁻²⁰. As previously described by Toro and colleagues¹⁷, local differences in cortical growth rate may serve as a complementary influence in cortical expansion-induced buckling models. Alternative perturbations to facilitate buckling were explored in *Supplementary Figure 1* and *Supplementary Table 1*.

8. p7 I9. What was Poisson's ratio for the fibres (in steady state)? One could imagine that shrinking due to Poisson's effect could trigger growth in fibre caliber to compensate stress, leading to a Poisson's ratio of ~0 instead of ~0.5 used for soft elastic tissue?

In this model, we consider fibers to possess similar elastic properties to the rest of the tissue, defined as a nearly incompressible hyperelastic material (bulk modulus 100 times greater than shear modulus), corresponding to a Poisson's ratio of 0.49. A clarification of the elastic Poisson's ratio has been added to Methods. This elastic behavior consistent with previous axon stretching experiments (for example, those described in Holland et al. 2015), which have shown measurable caliber change in fibers immediately after stretch (Poisson's ratio > 0 when considering the elastic response).

While we want to avoid confusing readers with conceptualization of "total" Poisson's ratio (including both elastic deformation and growth effects), we agree that this is an interesting point for exploration in future theoretical works.

9. p7 I13. In the model, under the action on stress, an element of volume will end up having fibres of a single orientation. However, in reality multiple directions often coexist (which is a well recognised problem for diffusion weighted imaging tractography). Van Wedeen et al. (2012) even proposed a conceptual model where white matter axons would always belong to 1 of 3 locally orthogonal orientations, based on DWI tractography data supported by histological data. How are these observations compatible with the model?

In our model, we chose to consider fibers in 3 (initially) orthogonal directions primarily because it represents the simplest model capable of reproducing the phenomenon of interest. However, we share in the reviewer's interest in the theory propose by Wedeen and colleagues. Current simulations and experimental data cannot directly refute or support whether this organization acts as a developmental mechanism. Additional research is needed to determine whether the orthogonal fiber assumption is appropriate as a definition, or only as a numerical and conceptual simplification.

In an attempt to remain neutral on this point, we have added a clarification of our motivation to Methods as follows:

The fiber volume fraction in each orthogonal direction (f_i) is a composite measure reflecting the number, length, and diameter of axons. ... More complex models could be developed to account for fiber scatter and the resulting shear growth contributions. However, such components are expected to have minor impact and have thus been omitted from previous simulations of stress-dependent subplate growth²².

10. p8 I7. In the model folding was triggered by making a small region grow 5% faster than the rest of the cortex (which reminds to some extent the differential expansion hypotheses of Welker, Ronan, Borrell,...). Was this necessary to trigger folding? Wouldn't the addition of a very small, homogeneous, geometric noise be sufficient? As far as I know, in most buckling models numerical noise is sufficient to trigger folding. Was the idea to control the pattern of

folding more precisely? What is the size of this small region? What is its shape? Was the rate of growth of this region 5% larger only at the beginning of the simulation or throughout the simulation?

As described in response to comment #7, we intentionally chose this perturbation as a biologically-plausible asymmetry to trigger folding on a controlled spherical geometry. We have added additional detail to the Methods describing this perturbation.

For simulations in which cortical growth was not uniform, a very small, localized increase in cortical growth (g_c^*) was added according to the function $g_c^* = 0.05 g_c (\cos \Theta)^{1000}$, where Θ describes the angle from edge to edge of the simulated curved surface (-90 to +90 degrees). As illustrated in Supplemental Fig. 1, the taper of this peak falls well within the bounds of an induced gyrus and did not affect the wavelength of folding. For simplicity, the 5% increase in growth for this region was maintained throughout the simulation.

We also considered the ellipsoidal case as an example of geometric asymmetry. Based on this and Reviewer 3's comments, the revised Supplement now includes a section (Supplementary Fig. 2) describing additional examples of local perturbation based on geometry and subplate organization.

11. p20 l3. Why cortical growth in the model is only tangential? Cortical thickness does increase through the period of cortical folding (likely due to the neuropil development, dendrites, axonal collaterals,...). In ferrets, for example, cortical thickness goes from <1mm at before folding (P4) to ~2mm at ~2 weeks (adult-like folding pattern). If growth is isotropic instead of just tangential such x2 increase in length, would translate to a x8 increase in volume, which is about the increase of brain volume observed within that period? Is the assumption that radial cortical growth is only the passive response to the pull of the white matter core and that all active growth is exclusively tangential?

Thickness is known to increase over the period of time in which folding occurs. However, the increase in thickness is very small compared with the increase in surface area in human brain development. For this reason, tangential-only growth has been justified and applied in many previous modeling studies of cortical folding, including the influential works of Xu et al. (2010), Bayly et al. (2013), Tallinen et al. (2014), Holland et al. (2015), Tallinen et al. (2016). This is now stated at the outset of our model description in the Results section.

Since increase in cortical thickness represents a relatively small component of total cortical growth, the cortex was not defined to grow in the radial (e_1) direction, consistent with past modeling studies^{17,22,23}.

Importantly, our model does induce an ~50% in cortical thickness despite application of only tangential cortical growth, as visible in figures illustrating model progression from initial to final time points. We have added a short description of this finding in the revised manuscript.

In the cortex, constrained expansion also induced tangential compression, resulting in elastic deformation and tension in the radial direction. In Fig. 2, this can be observed as an increase in thickness, with the cortical layer exhibiting an approximately 50% increase in thickness at $T=0.5$ and beyond. In both ferret and human cortices, an approximately 100% increase has been reported over a similar period^{30,31}. Thus, elastic deformation was sufficient to account for approximately half of the cortical thickening observed over this period of development.

While beyond the scope of the current manuscript, we note that stress-dependent growth could enhance this trend. More recent work by Maria Holland (Wang et al. BMMB 2020) suggests a possible role for mechanical feedback in cortical thickness:

<https://link.springer.com/article/10.1007/s10237-020-01400-w>

12. Because of the remodelling of the white matter fibres, I can't get a general idea of its overall elasticity, to be able to compare this model with previous models. In the context of the literature on buckling of bi-layers, does the present model correspond more to a wrinkling regime such as the models of Budday et al (2014), Bayly et al (2013) where the top layer is stiffer than the bottom one, or a creasing regime as strongly advocated by Tallinen, Mahadevan, et al (2014, 2016), where the elasticity in both tissues is similar?

In order to more clearly address this and other reviewer comments, Supplemental Fig. S1, has been moved to the main text and expanded as a new Results subsection (2.2). This includes a rough "translation" of the parameters / range of values used by Bayly et al. and Holland et al. in the context of our model.

- A nondimensionalized response ratio, R , was also considered to describe the subcortical fiber elongation rate (a) relative to cortical growth rate (g_c). $R=0$ corresponds to no fiber elongation, while larger R denotes faster fiber elongation relative to cortical growth rate. Bayly et al²² and Holland et al.²³ explored theoretical effects in the ranges of $R=1-40,000$ and $R=10-300$, respectively, under the assumption that this behavior applied to all subcortical tissue.
- Within the range that induced folding, faster fiber elongation (higher R) led to later onset of folding, and slower fiber elongation (lower R) led to earlier folding (Fig. 3b, left). An increase or decrease in the initial fiber volume fraction also resulted in later or earlier onsets of folding, respectively (Fig. 3b, right). These results are generally consistent with past models incorporating stress-dependent subcortical growth^{22,23}, which required slower rates to induce folding under similar boundary conditions ($R<32$) but approximated subcortical tissues to contain the maximum total fiber volume fraction ($f_1 + f_2 + f_3 = 1$).

In summary: While all layers of the current model possess the same elastic properties, stress-dependent axon elongation rates do alter the regime of folding. The faster axons are allowed to grow in response to stress, the "softer" the subcortical material behaves – corresponding to a wrinkling regime. On the other hand, as axon elongation rate approaches zero, we are left with a traditional hyperelastic material, with same stiffness as cortex, resulting in a creasing regime as advocated in the Tallinen models. As clearly stated in the new text, our model parameters and experimental data advocate for a wrinkling regime, rather than creasing regime.

Notably, fast fiber elongation also appeared to produce smoother, more sinusoidal folds at the onset of folding, while slower rates produced sharper, "cusped" sulci at the onset of folding (Fig. 3b). Tallinen and colleagues²¹ previously showed that sinusoidal buckling, or wrinkling, will only occur in models for which the cortex is stiffer than underlying subcortical layers. By contrast, if cortical stiffness is similar to (or softer than) the underlying tissue, a lower creasing threshold will be reached first, causing an immediate transition to sharp, self-contacting sulcal cusps that deepen over time. While the current model employs similar elastic properties for the cortex and subcortical layers, based on previously reported measures in the developing brain^{14,22,41}, inclusion of stress-dependent fiber elongation effectively increases compliance of the subcortical tissue. As shown in Fig. 2 ($R=60$), initially sinusoidal folding can lead to cusped sulcal morphology at later stages, as folds deepen in the post-buckling regime. This progression from shallow, relatively sinusoidal undulations to deep, sharp cusps within sulci is consistent with brain morphology over the course of development (Fig. 1a). Thus, current data supports an initial wrinkling instability, followed by deepening and sharpening of sulci into cusps over development.

13. p17 l10: "patterns *of* cortical"

Thank you for catching this typo – it has been corrected in the revised manuscript.

14. It would be very helpful to made the source code for the simulation available (as is for

example the case with the models of Toro et al. or Tallinen et al.). This could facilitate replication as well as future comparisons and extensions of the model.

Thank you for this comment. COMSOL files will be made available for future researchers, and this is noted in the Data Availability section. Model files have also been included with our resubmission for review (by anyone with access to a compatible version of COMSOL.)

R. Toro

Reviewer #3 (Remarks to the Author):

In this paper, the authors propose a new model that takes the volume fraction of axonal fiber into account. The model's outcomes – the predicted fiber organizations in the underlying white matter – are compared qualitatively against the high-resolution ex vivo diffusion tensor imaging of the developing rhesus macaque brain. The authors conclude that the proposed model supports a causative role for folding in the stereotyped fiber organization beneath gyri and sulci. While the model builds upon the authors' previous work (Bayly et al., 2013), this manuscript's novelty lies in incorporating the volume fraction of the evolving axonal fiber in the anisotropic material model.

While the paper is extremely interesting, I believe that there are several unanswered questions regarding the proposed constitutive model. Before this manuscript is ready for publication, the authors need to address the following concerns:

1. Compared to the evolution equation in their previous work (Bayly et al., 2013), the evolution equation proposed here contains the volume fraction f_i in front. This modification is briefly explained, but not fully justified. It is not immediately apparent to me that the growth rate (as opposed to the total volume change) should depend on the volume fraction of fibers. In addition to a more thorough justification of this governing equation, it would be helpful to further illustrate its effect under simple loading conditions. For instance, Fig. 2 could also contain graphs showing the total volume and the imposed stretch on the hypothetical tissue volume element with fibers.

We thank the reviewer for this important feedback and suggestion. In the revised manuscript, we have attempted to further justify and improve the clarity of our explanations. This has been facilitated by moving this portion of the text (and a revised Fig. 2) to Methods, allowing more space to fully describe these concepts.

Briefly, we note that fiber elongation rate does not depend on fiber volume fraction. However, the growth rate of the entire tissue does depend on the % of tissue containing growing fibers, since only fiber components are defined to be capable of growth.

According to Eq. 3, a block of tissue containing 100% fibers in direction i ($f_i=1$) will experience maximal growth for a given elongation rate and stress. Conversely, a block of tissue containing 0% fibers will experience no growth regardless of the applied stress.

We have extensively revised our theoretical description text (first subsection of Methods, p22-24) and Fig. 2 (now Fig. 8 in Methods) for clarity, in accordance with these suggestions as well as suggestions of the other reviewers, including more detailed explanations/justifications under simple conditions:

The consequences of these governing equations are illustrated in Fig. 8, considering a simple block of tissue subjected to sustained tension in one direction and zero stress in perpendicular directions. Based on histological studies suggesting that axonal fibers of the early subplate are sparse⁸, fiber volume fractions in this example are defined to account for only a small fraction of the overall tissue volume at the outset ($f_1 = f_2 = f_3 = 0.1$ at $t=0$). As fiber components parallel to the direction of tension elongate, growth is observed in the direction of tension (Fig. 8a). Since fibers are only defined to elongate, not to increase in cross-section (caliber), no tissue growth is observed in directions perpendicular to the applied tension. As fibers increase in length while maintaining their caliber, the corresponding fiber volume increases (Fig. 8b), and the relative volume fraction of fibers in this direction slowly approaches 1 as time approaches infinity (Fig. 8c). For any direction of sustained tension, the above equations also suggest an increase in total fiber volume fraction approaching 1 as time approaches infinity (Fig 8c).

2. While the volume fraction of axonal fiber changes in response to the stress, it does not in turn affect the stiffness of the material. Mechanical tests have shown that white matter has pronounced anisotropic and stretch-locking behaviors (Garcia-Gonzalez et al., 2018). The authors should consider this in their model, including a non-linear term in the mechanical free energy (Eq. 4) to capture this behavior.

We thank the reviewer for directing us to the work of Garcia-Gonzalez et al. While anisotropic and stretch-locking behaviors have been reported in brain tissues, these studies have focused on adult white matter tissue, in which axons are myelinated. Myelination has been shown to dramatically increase tissue stiffness. By contrast, the subplate is not myelinated at the stages considered in this study, and very few studies exist to test even the most basic mechanical properties of the subplate.

To the best of our knowledge, no studies to date have confirmed mechanical anisotropy in developing subcortical tissues, nor increased stiffness of unmyelinated axons relative to surrounding subplate tissues. This issue is now addressed in Methods as follows:

The same elastic properties were applied to all cortical and subcortical tissue components, including embedded fiber volume fractions. Anisotropic elastic properties could be incorporated by choosing an alternate form of W (Eq. 5), as previously described in adult brain tissue⁶¹. However, no studies to date have confirmed mechanical anisotropy in developing subcortical tissues nor increased stiffness of unmyelinated axons relative to surrounding subplate tissues. In the absence of this data, we assume isotropic elastic behavior, consistent with past models of cortical folding^{14,16,21,22}.

3. This study would be more compelling if the effect of various initial fiber organizations were investigated and illustrated. For instance, using a different coordinate system (such as Cartesian instead of spherical) for the initial fiber directions, or assigning slightly different initial fiber densities to certain direction(s) (instead of the predominantly radial orientation shown here). Are the findings robust to changes in initial alignment?

These are great points. Based on this suggestion, and related suggestions of Reviewer 1, we have added a new figure (Fig. 3) that explores the differences resulting from global changes in (1) stress-dependent fiber elongation rate, (2) initial fiber density, and (3) fiber orientations (isotropic and tangential instead of only considering radial). The results of different fiber densities are described in Results and copied below:

Within the range that induced folding, faster fiber elongation (higher R) led to later onset of folding, and slower fiber elongation (lower R) led to earlier folding (Fig. 3b, left). An increase or decrease in the initial fiber volume fraction also resulted in later or earlier onsets of folding, respectively (Fig. 3b, right).

Related to these effects, we note that initial fiber orientation also played a minor role in the observed morphology of folds (Fig 3c). When considering the same initial fiber volume fraction and fiber elongation rate, stronger radial organization produced sharper, deeper folds at $T=1$, while stronger tangential organization produced smoother, more shallow folds. In all cases, the stereotyped fiber organization with respect to folding – more radial beneath gyri and more tangential beneath sulci – was preserved.

The consideration of altered fiber density also led to valuable enhancements in our Discussion, in the context of polymicrogyria:

Finally, simulations suggest that a reduction in initial fiber density will also produce increased gyrification (Fig. 3b, right). This is consistent with experimental studies in which early disruption of subcortical connectivity (reduction of intact fibers) led to an overproduction of cortical folds^{5,52,53}. Alternatively, as suggested by previous work¹⁷, disruption of connectivity may alter a complex array of developmental mechanisms, such as cortical growth, which could also alter the observed pattern of folding.

We also conducted a detailed analysis of the influence resulting from local, rather than global, variations in initial fiber organization in the Supplementary Fig. 2 and Table 2. These results illustrate that local differences in initial fiber density or orientation can dictate whether a gyrus or sulcus is induced in specific locations. In most cases, initial alignments are complementary to the stress-induced remodeling described in our study: a local increase in radial orientation or fibers leads to a gyrus, while a local increase in tangential orientation or fibers leads to a sulcus. As described in Discussion:

Previous simulations of cortical expansion-induced folding have illustrated the potential for pre-existing subplate orientations to bias the location and direction of folds²³, and this general finding is confirmed in the present model (Supplementary Fig. 2 and Table 2). However, such patterns are not observable at the onset of folding, nor are they necessary to explain the observed link between folding and subplate organization. Fiber remodeling induced by folding stresses may even be sufficient to override weak heterogeneities in subplate orientation (Supplementary Fig. 2).

4. In Figure 4d, fibers appear to be highly aligned under both the gyral crown and sulcal fundi, but there is an observable difference in FA value between the two regions. Similarly, in the macaque brains, the FA value does not change significantly in time or vary significantly between gyri and sulci. This difference should be discussed further.

The reviewer has made an excellent point. We have revised our manuscript to recognize that there are additional determinants of water diffusion anisotropy beyond the width of the orientation distribution of fibers within a voxel, which is the property reflected in FA*. Such factors include the density of fibers (at low density, FA will be low even if all fibers are oriented in a similar manner) and maturation of the fiber bundle (such as myelination). Nevertheless, there are examples in the literature where relatively high FA within gyral developing white matter is apparent, and we refer the reader to an example of the G135 rhesus macaque. The specific text addressing this issue is:

Notably, the magnitude of diffusion anisotropy, reflected in FA (Fig. b, inset), differs from the magnitude of anisotropy in fiber orientation distributions, reflected in FA* (Fig. d), with the latter exhibiting a pattern of higher values within gyral subplate relative to tissue bordering sulcal gray matter. By contrast, diffusion anisotropy in the G110 subplate is uniformly low. There are several possible reasons for this difference,

such as a dependence of FA on fiber density and maturation state of myelin⁴³. Indeed, at later stages of development, higher FA is observable within developing gyral white matter compared to sulcal regions (e.g., in the G135 brain as can be observed in Fig. 7b of ⁴³).

5. The choice of exponential cortical growth, while based on data from macaque development, is unusual. Most, if not all, of the studies used for comparison in this manuscript used linear growth. Considering that a major point is comparisons between axonal growth rates in these studies, this presents a significant problem, as axonal and cortical growth rates can really only be understood relative to each other. This choice should be justified further, and its effect on the relationship between axonal and cortical growth rates should be discussed and, if possible, quantified.

We appreciate the Reviewer raising this point and suggesting that we provide further justification. We have added additional justification to the Methods of the revised manuscript. We have also added a comparison to linear growth rate in Supplementary Fig. 3.

In short, the linear cortical growth form corresponds to faster growth at earlier time points and slower growth at later time points. This leads to an earlier onset of folding (for the same axon elongation rate and total cortical expansion over the given period of time). While fold morphology and the time at which folds appear is altered, the important trends relative to folds are unchanged.

- Methods subsection “Finite element model”: Cortical growth rate, g_c , was estimated from an exponential fit of total surface area change observed during early rhesus macaque development²⁹. Previous models have considered both linear^{16,22,23} and nonlinear¹⁷ cortical growth. Much remains unknown from a mechanistic perspective. However, previous work has referred to a “slow” linear growth phase associated with cell migration and somal intercalation, followed by a “fast” phase that is coincident with cytoarchitectural differentiation⁶³.
- Methods subsection “Parameter determination for stress-dependent fiber elongation”: To bring past measures and models into a consistent framework, we also defined $R = G^{\text{axon}}/g_c$ as a nondimensionalized parameter representing the subcortical fiber response rate relative to the cortical growth rate. Under the assumption of an incompressible material undergoing small deformations and uniaxial tension, the stress-strain relationship for model fibers was again approximated as $\sigma = E\epsilon$, with $E = 3\mu$ approximating the Young’s modulus of model fibers in this case. Thus, for a given value of a , $G^{\text{axon}} = 3a\mu$ and $R = G^{\text{axon}}/g_c = 3a\mu/g_c$. This nondimensionalization approach is similar to the approaches described by Bayly et al²² and Holland et al.²³, though ratios of these studies were related to a linear cortical growth rate. Models in this study focused on the ratio $R=60$ (Figs 2, 4, 6, Supplemental Figs. 1-3), and behavior across a range of values was explored in Fig. 3. Comparisons relative to linear cortical growth rate are explored in Supplementary Fig. 3.

Additionally, I believe the paper would be strengthened by addressing the following comments:

1. The term 'viscoelastic growth' is used in several places to refer to the behavior of the subplate, and it is not clear what is meant by this. Growth represents the addition of new mass while viscoelasticity is the result of microstructural changes. While the resulting behavior appears similar in some contexts, the two processes are not the same, and this distinction should be made more clearly to avoid confusing readers.

We thank the reviewer for this comment. We had attempted to use “viscoelastic” as a more accessible term than “stress-dependent growth”, thinking that this may be more familiar to a

broad scientific audience. However, we agree that it is less precise and could lead to confusion. We have replaced all references to “viscoelastic” behavior with a more accurate descriptions of stress-dependent/stress-induced growth/elongation or mechanical feedback.

2. The authors explain their calculations of the stress-dependent fiber elongation parameter at length. However, consider that the experiments analyzed in Holland et al. 2015 include stretches of almost 800% in an hour, which is hardly physiological. Assuming that most experiments rely on similarly aggressive stresses and stretches, this might suggest that parameters based on *in vitro* experiments represent upper bounds of what axons are capable of, rather than the quantification of physiological behavior.

We agree with the Reviewer that *in vitro* experiments may represent the upper bound of what axons are capable of. We have supplemented our original Discussion point on this topic with the points below:

By applying axon elongation rates to only fiber components, a relatively small proportion of early subcortical tissue, net growth of the subplate was reduced to a level that facilitates folding. Still, we note that only rates at the lower end of the reported range were capable of producing a folded cortex. This result may suggest a slower response of neurites under *in vivo* conditions during the period of folding, with *in vitro* experimental conditions (including relatively rapid stretches compared to those anticipated during brain development) representing the upper bound of axon elongation behavior.

Notably, while initial stretch applied to fibers in studies such as Holland et al. was beyond the physiological levels, our parameter of interest (fiber elongation rate in response to stretch) was calibrated by how fast the axons lengthened in response over the entire experimental period (2-30 hours). Fiber elongation rate calibrated by Holland et al. 2015 was on the lower end of reported measures, 0.08 *per unit stretch* per hour. (A sustained stretch of 100%, doubling the length of the axon at rest, led to 8% axon growth per hour.) While this rate seems more reasonable, we agree that even this may be faster than the expected in the developing brain.

At the request of other reviewers, detailed characterization of this parameter are now explored in Fig. 3 and the accompanying Results text (an expansion of the material formerly provided as Supplement). We note that, even at lower fiber elongation rates, the same general trends are observed with respect to fiber reorganization beneath gyri and sulci.

3. The evolution equation (Eq. 2) also allows for tissue shrinkage when $\sigma_i < \sigma_0$, or when the tissue is in compression. This is alluded to in lines 11 and 12 but should be made explicit.

We have added an explicit description to the Methods regarding this point.

4. The numerator and denominator in Eq. 3 appear to be (unitless) volume ratios, not volumes. Perhaps the second term would be better written as $J_{\text{fibers},i} / J_{\text{total}}$.

Thank you for this suggestion. We agree that it is clearer to describe these unitless ratios and have added clarification in the updated text. However, to avoid potential confusion regarding this term and the elastic Jacobian, J , we have maintained the term V , which is also used to describe volumes in our updated Fig. 2 (now Fig. 8).

References

PV Bayly, RJ Okamoto, G Xu, Y Shi, and LA Taber. A cortical folding model incorporating

stress-dependent growth explains gyral wavelengths and stress patterns in the developing brain. *Physical biology*, 10(1):016005, 2013.

Daniel Garcia-Gonzalez, A Jerusalem, Sara Garzon-Hernandez, R Zaera, and A Arias. A continuum mechanics constitutive framework for transverse isotropic soft tissues. *Journal of the Mechanics and Physics of Solids*, 112:209–224, 2018.

Reviewer #4 (Remarks to the Author):

The authors propose that stress induced by cortical growth and folding might explain the white matter configurations seen in the underlying white matter. In particular, they argue that stress-induced fibre growth might explain the tangential orientations in the white matter underlying sulci and the radial orientations in the white matter underlying gyral crowns. These patterns are shown in ex-vivo macaque DTI data for comparison. This is an interesting proposal, however I am confused about the underlying mechanics.

Model queries:

1. Could the authors clarify what is actually happening during the initial tissue stretch and the subsequent fiber elongation phase shown in Figure 2. How can the tissue stretch without the axons within it elongating (or breaking)? And if the fibres elongate during the tissue deformation phase, what is actually the growth in the second phase? The illustration in Figure 2 seems to suggest that during the "fiber elongation" phase it is actually the axonal diameters that change, which does not appear in line with my (admittedly very cursory reading) of the papers describing fiber elongation under stress. It would be very helpful to spell out explicitly what the model implies for what happens with the axonal length, axonal diameter, and number of axons under stress, keeping in mind that the tissue cannot deform without axonal elongation/shrinkage.

We thank the reviewer for bringing up this issue of clarity. Based on this comment, as well as comments from the other reviewers, we have refined Fig. 2 (now Fig. 8) to more clearly illustrate the phenomenon described in our model. While tissue stretch and elongation represent two distinct processes, they occur simultaneously and continuously over the course of development, such that the elastic tissue stretch at any time remains low, but the cumulative effect of sustained stretch induces extreme elongation. The original figure attempted to illustrate these two processes sequentially, using an exaggerated cartoon, for a single time step. The revised figure describes this as a continuous, iterative process.

In the revised manuscript, we have substantially refined our explanation (pages 22-24). This includes a note that the current model depends only on fiber volume fraction, as well as elaboration on model behavior for simple cases. As now described in Methods:

- The fiber volume fraction in each orthogonal direction (f_i) is a composite measure reflecting the number, length, and diameter of axons.
- The consequences of these governing equations are illustrated in Fig. 8, considering a simple block of tissue subjected to sustained tension in one direction and zero stress in perpendicular directions. Based on histological studies suggesting that axonal fibers of the early subplate are sparse⁸, fiber volume fractions in this example are defined to account for only a small fraction of the overall tissue volume at the outset ($f_1 = f_2 = f_3 = 0.1$ at $t=0$). As fiber components parallel to the direction of tension elongate, growth is observed in the direction of tension (Fig. 8a). Since fibers are only defined to elongate, not to increase in cross-section (caliber), no tissue growth is observed in directions perpendicular to the applied tension. As fibers increase in length while maintaining their caliber, the corresponding fiber volume increases (Fig.

8b), and the relative volume fraction of fibers in this direction slowly approaches 1 as time approaches infinity (Fig. 8c).

2. The model does not appear to take into account extra-cellular space. While stretching tissue in one dimension would reduce the density of cells and axons not aligned with the stretch, the extra-axonal space will increase as the tissue volume increases. This would prevent the fibre volume fraction from going to one (Figure 2). This assumption of no extra-cellular space greatly influences the total fibre density increase seen in Figure 3, which would instead remain roughly constant if the extra-axonal space was extra-cellular rather than cellular.

In our model, the volume fraction, f_c , was defined to account for both cells and extracellular space (extracellular matrix and water). However, we appreciate the points that extracellular space persists in the adult brain (15-20%) and the total fiber volume fraction never truly reaches 1, as illustrated in our theoretical example.

Importantly, we note that the total fiber volume fraction remained well below 0.75 throughout our folding simulations, as illustrated in Fig. 2c (formerly Fig. 3). For this reason, we feel that addition of complicating equations/parameters is not appropriate for the main analysis presented here. However, we have included a Supplemental analysis to illustrate the theoretical effects of allowing extra-axonal space to increase alongside fiber volume, as referenced in the revised Results and Methods. Notably, at deposition rates that facilitate a final fiber volume fraction consistent with observations in adult (15-20%), we did not observe differences that affect the conclusions of this paper.

Results

- To match the low fiber content and weak radial orientation observed in early subcortical layers of gyrencephalic species³², radial and tangential fiber volume fractions were initially set at $f_{1,0}=0.12$ and $f_{2,0}=f_{3,0}=0.1$, respectively, with remaining non-fiber volume fraction (cell bodies, extracellular matrix) defined by $f_{c,0}=1-f_{1,0}-f_{2,0}-f_{3,0}$.
- Throughout the primary analysis presented, non-fiber components were considered to be non-growing. However, growth of non-fiber components was also explored, resulting in similar patterns (Supplementary Fig. 2).

Methods:

- For any direction of sustained tension, the above equations also suggest an increase in total fiber volume fraction approaching 1 as time approaches infinity (Fig 8c). Notably, fiber volume fraction in the adult white matter is known to be closer to 0.8, with extracellular matrix (ECM) accounting for approximately 15-20% of tissue volume^{59,60}. To define a maximum fiber volume fraction <1 , growth behavior may be modified to produce a set volume of ECM alongside new fiber volume, as described in *Supplementary Information*.

3. The result of the simulations seem to imply that the increase in the white matter's fractional anisotropy during development can be explained by tissue deformations, without the traditional explanations of new axon formation and myelination. Do the authors actually think this is the case? If so, it would be good to actually discuss this surprising result in the discussion.

This point was also raised by Reviewer 1. We have added text to clarify that we consider stress-induced growth to be only one factor that influences white matter development, but it is not exclusive. As stated in the revised text:

Results

Notably, the magnitude of diffusion anisotropy, reflected in FA (Fig. b, inset), differs from the magnitude of anisotropy in fiber orientation distributions, reflected in FA* (Fig. d), with the latter exhibiting a pattern of higher values within gyral subplate relative to tissue bordering sulcal gray matter. By contrast, diffusion anisotropy in the G110 subplate is uniformly low. There are several possible reasons for this difference, such as a dependence of FA on fiber density and maturation state of myelin⁴³. Indeed, at later stages of development, higher FA is observable within developing gyral white matter compared to sulcal regions (e.g., in the G135 brain as can be observed in Fig. 7b of ⁴³).

Discussion

It is important to acknowledge that the current model only approximates growth by focusing on specific developmental mechanisms, such as tension-induced fiber elongation and cortical expansion. For example, it does not account for axon growth resulting from other biological mechanisms such as well-established molecular axon guidance factors⁵⁴. Indeed, prior to cortical folding, afferent axons from cortical and subcortical structures form synapses within the subplate, and efferent axons project from the cortex based on previously described non-mechanical cues. ...

Minor comments:

1. When introducing the mechanical stress σ it would be very helpful to mention that this is related to the elastic deformation tensor F^* through equations given in the methods section, so the reader knows where to find the remaining equations.

Thank you for this suggestion. We have added an explanation to this effect.

2. It would be good to mention that the layer selected in Figure 6 is supposed to represent the "superficial subplate" in the caption or the main text rather than only in the methods section.

Thank you for this suggestion. A note has been added to the caption of Figure 5.

3. The fibre orientations in Figures 4-6 are hard to see. I would suggest to make the figures bigger and/or color code the fibre orientations based on how radial orientations. The latter would make it a lot easier to see the main point of these figures at a glance.

Thank you for this suggestion. We have increased the size to improve visualization of fiber orientations.

4. Page 13; line 6: insert "of" in "features subcortical fiber organization"

Thank you, this has been corrected.

5. Page 13; line 27: "FA increased" should be "increased FA"

Thank you, this has been corrected.

6. Please define J^* in equation 4

Thank you for pointing out this omission. A definition has been added.

REVIEWERS' COMMENTS

Reviewer #1 (Remarks to the Author):

I was very impressed with how well you addressed all of the reviewer concerns. You produced one of the best response/rebuttal letters I've ever read, and I think your revisions have made the manuscript more readable, largely because the highly technical parts are now more separated from the general conceptual matters. Because the other reviewers are clearly more expert at the technical aspects of your work, I will leave it to them to ensure that no mistakes have crept in, but I urge you also to triple check your revisions. Finally, I just want to point out a missing "on" on page 3, line 16; I also suggest you write "applying axon elongation rates only to fiber components" on page 18, line 25.

Reviewer #2 (Remarks to the Author):

The authors have answered all my questions satisfactorily. I don't have further comments, and in my opinion the paper should be accepted for publication.

Reviewer #3 (Remarks to the Author):

This paper has been much improved in revisions. Upon rereading, and in light of the authors' modifications, a few really important points stood out to me. These are, at least in the first two cases, observations I had perhaps made but not articulated even to myself, and I believe the field will be better for having them made explicitly:

1. stress-dependent fiber elongation effectively decreases subcortical stiffness
2. sinusoidal and creased folding are not mutually exclusive; the former may morph into the latter
3. fiber fraction can be a bridge between in vitro experiments, where only axons are stretched, and simulations of whole brain tissue.

I have only a few remaining comments and suggestions, and hope the paper can be accepted and published within a reasonable timeline.

4. Fig. 8b is based on six orders of magnitude of growth, which seems to be overdone and absurdum
5. I don't quite agree with the argument that stress-dependent subcortical growth is exponential; this is only true under constant stress, which is not realistic. (Perhaps some regions are under constant stress, but certainly not the tissue as a whole.) I do, however, understand that it is difficult to meaningfully compare two quantities such as subcortical and cortical growth which are obviously related, but equally obviously differ in significant ways.
6. The tables need much more complete captions; e.g. Supplementary Table 1 should explain the meaning of asterisks and bold text.
7. Fig. 3 is a bit confusing, as both rates and times are being changed simultaneously. How were the times chosen?

Very minor edits:

8. missing minus sign in p7 line 11
9. missing 1 in units of Gaxon in Table 1
10. perfusion-fixed is missing a hyphen (p28)
11. the confusing term "pre-folded" has still been used in the paper

Reviewer #4 (Remarks to the Author):

Thanks to the authors for their careful consideration of my comments. The changes made in the Methods sections have (at least for me) greatly clarified these simulations of how cortical growth affects the underlying fibre density and orientation. The results are very interesting, in particular after the inclusion of a look at how the results depend on model parameters as suggested by other reviewers.

I have no further comments, except for noting that there is a minus sign missing in the equation for $f_{c,0}$ on page 7, line 11.

We would like to again thank the reviewers for their positive remarks and suggestions. Minor changes to the revised manuscript are outlined point-by-point below.

New changes to the main text, including those based on editorial comments/formatting requirements, are highlighted in red in the revised manuscript document. (Since the Supplement must be submitted in final form, changes to that content are only described below.)

Reviewer #1 (Remarks to the Author):

I was very impressed with how well you addressed all of the reviewer concerns. You produced one of the best response/rebuttal letters I've ever read, and I think your revisions have made the manuscript more readable, largely because the highly technical parts are now more separated from the general conceptual matters. Because the other reviewers are clearly more expert at the technical aspects of your work, I will leave it to them to ensure that no mistakes have crept in, but I urge you also to triple check your revisions. Finally, I just want to point out a missing "on" on page 3, line 16; I also suggest you write "applying axon elongation rates only to fiber components" on page 18, line 25.

We thank the reviewer for their positive remarks regarding our response to initial reviewer comments. We appreciate the urging to triple check our revision, which we have done. We have corrected the missing "on" on page 3 and adjusted the phrasing on page 18 as suggested.

Reviewer #2 (Remarks to the Author):

The authors have answered all my questions satisfactorily. I don't have further comments, and in my opinion the paper should be accepted for publication.

We thank the reviewer for their approval of our revised manuscript.

Reviewer #3 (Remarks to the Author):

This paper has been much improved in revisions. Upon rereading, and in light of the authors' modifications, a few really important points stood out to me. These are, at least in the first two cases, observations I had perhaps made but not articulated even to myself, and I believe the field will be better for having them made explicitly:

1. stress-dependent fiber elongation effectively decreases subcortical stiffness
2. sinusoidal and creased folding are not mutually exclusive; the former may morph into the latter
3. fiber fraction can be a bridge between in vitro experiments, where only axons are stretched, and simulations of whole brain tissue.

Thank you for your positive comments – we greatly appreciate the reviewers' input, which prompted us to flesh out and emphasize these points.

I have only a few remaining comments and suggestions, and hope the paper can be accepted and published within a reasonable timeline.

4. Fig. 8b is based on six orders of magnitude of growth, which seems to be overdone ad absurdum

While we agree that this is an unnecessary order of magnitude in terms of illustrating growth (change in volume). However, this length of time (and absurd magnitude of growth) is necessary to illustrate the theoretical asymptotic approach toward a volume fraction of 1. We recognize that this extreme growth is not biologically plausible, but still feel this theoretical/conceptual illustration of our new constitutive equations is useful. Thus, we have kept limits the same in Fig. 8b, but we have revised the caption to subtly reframe the focus of this panel.

Figure 8. Theoretical growth and remodeling in response to sustained tension. (a) Schematic illustration of tissue elements with embedded fibers in initial configuration (left), after a short period of stretch and fiber elongation (middle), and after an extended period of stretch and fiber elongation (right). Tissue blocks are not shown to scale since volume increase becomes dramatic over an extended period of stretch. (b) Theoretical behavior of a tissue element subjected to sustained tension in direction \mathbf{e}_1 as time approaches infinity. Top graph illustrates the total change in volume, starting from a nondimensionalized total tissue volume (V_{total}) of 1 at the initial time point. V_1 represents volume associated with fibers aligned in the direction of tension (\mathbf{e}_1), while V_2 and V_3 represent the volume associated with fibers perpendicular to tension (\mathbf{e}_2 and \mathbf{e}_3), and V_c represents the volume associated with cells and extracellular matrix. Vertical axis is log scaled to facilitate visualization, as total volume begins to increase dramatically (approximately exponentially) at later time points. Bottom graph illustrates the effect on volume fraction of each tissue component, starting from an initial fiber volume fraction of 10% in each direction. When tension and fiber growth are allowed to continue indefinitely, fiber volume fraction approaches 1 while the volume fraction of non-fiber components (f_c) approaches 0.

5. I don't quite agree with the argument that stress-dependent subcortical growth exponential; this is only true under constant stress, which is not realistic. (Perhaps some regions are under constant stress, but certainly not the tissue as a whole.) I do, however, understand that it is difficult to meaningfully compare two quantities such as subcortical and cortical growth which are obviously related, but equally obviously differ in significant ways.

We agree that exponential subcortical growth (as shown in Fig. 8) is only true under constant stress. Under more realistic conditions, such as the finite element models in this study, stress-dependent growth would be expected to relieve stress over time, and subcortical growth would taper as a consequence of reduced tension. In this way, an exponential form, in terms of constitutive equations, can still accurately capture non-exponential growth behavior.

We stress that Fig. 8 has been provided as a conceptual illustration. We have added several instances of the term "theoretical" to stress this in the manuscript.

- The theoretical consequences of these governing equations are illustrated in Fig. 8, considering a simple block of tissue subjected to sustained tension in one direction and zero stress in perpendicular directions.

- **Figure 8. Theoretical growth and remodeling in response to sustained tension.**

6. The tables need much more complete captions; e.g. Supplementary Table 1 should explain the meaning of asterisks and bold text.

Thank you for pointing out this omission. The special cases in Supplementary Table 1 are now denoted as shown below:

Supplementary Table 1. Predicted fold morphology above localized variation in subplate fiber distribution

Local fiber volume fraction relative to surrounding subplate	Initial predominant fiber orientation	High curvature model prediction	Low curvature model prediction
Higher or Same	$1 > 2 = 3$	Gyrus	Gyrus
	$2 > 1 > 3$	Sulcus	Sulcus
	$3 > 1 > 2$	Gyrus* ⁺	Sulcus* ⁺
Lower or Same	$2 = 3 > 1$	Sulcus ⁺	Sulcus ⁺
	$1 > 3 > 2$	Gyrus	Gyrus
	$1 > 2 > 3$	Sulcus*	Gyrus*

*predicted direction of folding depends on initial model curvature

⁺stress-induced remodeling deviates from the initially predominant fiber orientation

7. Fig. 3 is a bit confusing, as both rates and times are being changed simultaneously. How were the times chosen?

Times were chosen to illustrate morphology and fiber volume fractions after the onset of folding, which occurred later for higher rates/fiber densities. We appreciate the reviewer raising this confusion. We have added a statement to the caption to clarify this point:

(b) Rate of fiber elongation and initial fiber density similarly alter the time course and morphology of cortical folding. For each case, simulation time (T) was chosen to illustrate morphology and fiber volume fractions at a similar stage of fold formation.

Very minor edits:

8. missing minus sign in p7 line 11

Thank you, this omission has been corrected.

9. missing 1 in units of Gaxon in Table 1

This omission has been corrected.

10. perfusion-fixed is missing a hyphen (p28)

This omission has been corrected.

11. the confusing term "pre-folded" has still been used in the paper

We apologize for failing to catch this instance in the Fig. 7 caption. We have replaced the term “pre-folded” with “G85”.

Reviewer #4 (Remarks to the Author):

Thanks to the authors for their careful consideration of my comments. The changes made in the Methods sections have (at least for me) greatly clarified these simulations of how cortical growth affects the underlying fibre density and orientation. The results are very interesting, in particular after the inclusion of a look at how the results depend on model parameters as suggested by other reviewers.

I have no further comments, except for noting that there is a minus sign missing in the equation for $f_{c,0}$ on page 7, line 11.

We thank the reviewer for their approval of our revised manuscript, and we appreciate this catch of a missing minus sign on page 7. This error has been corrected in our final manuscript.